# Volatility measurement of atmospheric submicron aerosols in an urban atmosphere in southern China

Li-Ming Cao[1], Xiao-Feng Huang[1], Yuan-Yuan Li[1], Min Hu[2], Ling-Yan He[1]

[1]Key Laboratory for Urban Habitat Environmental Science and Technology, School of Environment and Energy, Peking University Shenzhen Graduate School, Shenzhen, 518055, China
[2]State Key Joint Laboratory of Environmental Simulation and Pollution Control, College of Environmental Sciences and Engineering, Peking University, Beijing, 100871, China

*Correspondence to*: Xiao-Feng Huang (huangxf@pku.edu.cn)

**Abstract.** Aerosol pollution has been a very serious environmental problem in China for many years. The volatility of aerosols can affect the distribution of compounds in the gas and aerosol phases, the atmospheric fates of the corresponding components and the measurement of the concentration of aerosols. Compared to the characterization of chemical composition, few studies have focused on the volatility of aerosols in China. In this study, a Thermo-Denuder - Aerosol Mass Spectrometer(TD-AMS) system was deployed to study the volatility of non-refractory submicron particulate matter ($PM_1$) species during winter in Shenzhen. To our knowledge, this paper is the first report of the volatilities of aerosol chemical components based on a TD-AMS system in China. The average $PM_1$ mass concentration during the experiment was $42.7 \pm 20.1$ μg m$^{-3}$, with organic aerosol (OA) being the most abundant component (43.2% of the total mass). The volatility of chemical species measured by the AMS varied, with nitrate showing the highest volatility, with a mass fraction remaining (MFR) of 0.57 at 50 ℃. Organics showed semi-volatile characteristics (the MFR was 0.88 at 50 ℃), and the volatility had a relatively linear correlation with the TD temperature (from the ambient temperature to 200 ℃), with an evaporation rate of 0.45 % ·℃$^{-1}$. Five subtypes of OA were resolved from total OAs by positive matrix factorization (PMF) for data obtained under both ambient temperature and high temperatures through the TD, including a hydrocarbon-like OA (HOA, accounting for 13.5%), a cooking OA (COA, 20.6%), a biomass burning OA (BBOA, 8.9%) and two oxygenated OAs (OOA): a less-oxidized OOA (LO-OOA, 39.1%) and a more-oxidized OOA (MO-OOA, 17.9%). Different OA factors presented different volatilities, and the volatility sequence of the OA factors at 50 ℃ was HOA (MFR of 0.56) > LO-OOA (0.70) > COA (0.85) ≈ BBOA (0.87) > MO-OOA (0.99), which was not completely consistent with the sequence of their O/C ratios. The high volatility of HOA implied that it had a high potential to be oxidized to secondary species in the gas phase. The aerosol volatility measurement results in this study provide useful parameters for the modelling work of aerosol evolution in China and are also helpful in understanding the formation mechanisms of secondary aerosols.

## Introduction

Atmospheric aerosol pollution has important impacts on climate change, visibility and human health (Bohnenstengel et al., 2014; Baklanov et al., 2015). Aerosols can be emitted, naturally or anthropogenically, by primary sources or produced by secondary chemical reactions from gaseous precursors (IPCC, 2013). Volatility is one of the most important properties of aerosols, as it can determine the gas-particle phase partitioning of aerosols directly. The saturation vapour pressure, which is affected by temperature and vaporization enthalpies as described by the Clausius-Clapeyron equation, is the main factor that

dominates the gas-particle phase partitioning of compounds. Volatility can be determined by the saturation vapour pressure or saturation concentration, while the deposition rates of aerosols for wet or dry deposition are greatly influenced by the phase of aerosols (Bidleman, 1988). The chemical mechanism and reaction rates of gas, liquid and heterogeneous reactions can also result in a great difference because of the phase, the concentration and the lifetime of aerosols can also be influenced (Huffman et al., 2009a). The volatility of organic aerosol (OA) also contributes greatly to the uncertainty in predicting the atmospheric

aerosol concentration (Donahue et al. 2006; Pankow and Barsanti 2009). Since OAs contribute 30-80% of total aerosol mass according to previous studies (Hallquist et al., 2009; Zhang et al., 2007a), further research regarding the volatility of atmospheric aerosols, especially organic aerosols, is very important in the understanding of aerosol characterization and source apportionment. Some previous studies examined several kinds of OA emission sources, including traffic emissions, combustion sources and the oxidation of primary OA (POA), and showed differences in volatility according to their different

compositions (Huffman et al., 2009a; Xu et al.,2016a; Paciga et al., 2016).

    A thermo-denuder (TD) is a device that is widely used to estimate aerosol volatility distributions (Wehner et al., 2002; An et al., 2007; Huffman et al., 2008; Xu et al., 2016a). The TDs designed by Burtscher et al. (2001) and Wehner et al. (2002) are typically operated under temperatures higher than 200 ℃ and have average residence times from 0.3 to 9 s, focusing on very low volatility species. An et al. (2007) and Huffman et al. (2008) developed TDs with longer residence times to make them

more suitable for measuring the volatility of semi-volatile organic aerosols. The combined TD and Aerodyne Aerosol Mass Spectrometer (TD-AMS) system was firstly applied in ambient study by Huffman et al. (2008) to quickly characterize the volatility of chemically-resolved ambient aerosol in a field campaign, and the temperature profiles, particle losses and key factors affecting the results were discussed. Huffman et al. (2009a) then measured the volatility of OA from different sources, including biomass-burning OA, meat-cooking OA, trash-burning OA, and chamber SOA formed from α-pinene and gasoline

vapours, and found semi-volatility for all the OAs, which is opposite to the previous atmospheric models that only regarded POAs as non-volatile species. Huffman et al. (2009b) also analyzed the positive matrix factorization (PMF) results based on the TD-AMS data and demonstrated that all types of OA should be regarded as semi-volatile species in the models. Lee et al. (2010) measured the volatility of aerosols with two different residence time sets and suggested that longer residence time was required to constrain the variation of OA volatility at lower concentrations. Obviously, OA volatilities, especially for different

OA types, are still quite uncertain and need more ambient measurements to constrain.

Aerosol pollution has been one of the most important air quality problems in China. Many studies focused on aerosol source apportionment and chemical and physical properties have been carried out in most of the regions in China, especially in the Yangtze River Delta Region, the Pearl River Delta Region, the Beijing-Tianjin-Hebei Region and northwest China in the past few years (Huang et al 2010, 2012, 2013; He et al., 2011; Xu et al., 2014; Li et al., 2017). However, the volatility of aerosols is rarely researched in China currently. Cheung et al. (2016) studied the aerosol volatility in Guangzhou, China, based on the volatility tandem differential mobility analyzer (VTDMA) measurement, and found that non-volatile organic aerosol may contribute significantly to the non-volatile residuals. Also using a VTDMA, Nie et al. (2017) studied the volatility of aerosol humic-like substances (HULIS) in Nanjing, China, and figured out that the interaction between HULIS and ammonium sulfate tended to decrease the volatility of organic aerosols. In this paper, the TD-AMS system was firstly deployed to determine the volatilities of non-refractory $PM_1$ species in China. With the high-resolution mass spectra of organics, the volatilities of OAs from different sources and their implications for organic aerosol ageing were also explored.

**Experimental methods**

**2.1 Sampling site description**

Shenzhen (113.9 °E, 22.6 °N) is located on the southeast coast of China, in the southeast corner of the Pearl River Delta region with Hong Kong neighbouring to the south and Dongguan (a famous industrial city in China) to the north. The climate in Shenzhen is a subtropical oceanic climate that is deeply influenced by monsoon. The sampling site was located on the campus of Shenzhen Graduate School, Peking University in the western urban area of Shenzhen. The area surrounding the sampling site was mostly covered by subtropical plants, and there was only a local road that was approximately 100 m away, which can be regarded as an anthropogenic emission source. The measurement was taken from 31 December 2014 to 23 January 2015 in winter, which is the season with the highest air pollution due to regional transportation from northwest and northeast China. The average ambient temperature was 15.9 ±4.2 ℃, and the relative humidity was 62.9 ±17.5 %. The wind was mostly from the northeast and northwest with an average speed of 0.8 ±0.7 m/s.

**2.2 Instruments and methodology**

The aerosol volatility measurements were conducted with a TD – AMS system. The thermo-denuder and the high resolution time-of-flight aerosol mass spectrometer (HR-ToF-AMS, referred to hereafter as AMS) were both manufactured by Aerodyne Research Inc., US. The principle theory of an AMS can be found in previous studies (DeCarlo et al., 2006; Canagaratna et al., 2007). The AMS was set into two sampling ion optical modes: the V mode with unit mass resolution (UMR) was used for quantification of the UMR mass concentration and size distribution of the non-refractory species (including organics, sulfate, nitrate, ammonium, and chloride); the W mode was used to obtain the high-resolution mass spectra (~3000 m/Δm), and the ions with m/z up to 170 could be separated properly in this study. The calibrations were conducted at the beginning and end of the campaign with the method described previously (Jayne et al., 2000; Jimenez et al., 2003; Drewnick et al., 2005),

including the inlet flow rate, ionization efficiency calibration (IE), and particle size calibrations. The relative ionization efficiencies (RIE) used in the study were 1.2 for sulfate, 1.1 for nitrate, 1.3 for chloride, 1.4 for organics, and 4.0 for ammonium, respectively (Jimenez et al., 2003). The TD used in this experiment is based on the design of Huffman et al. (2008). It consists of two parts: the heating section and the denuder section. The stainless steel heating section is 22.25 inches (56.5 cm) in length with a 1-inch OD (2.5 cm) and a 0.875-inch ID (2.2 cm), wrapped with three fibreglass-coated heating tapes. The heating section is then joined to a 22-inch (56 cm) denuder section. The denuder section is filled with activated charcoal at room temperature to adsorb the gas phase species evaporated from particles. The temperature in the heating section was set at 48, 95, 143 and 192 ℃ to make the real temperature at the centreline, measured with a thermocouple, reach 50, 100, 150 and 200 ℃, respectively. The full configuration of temperatures in the TD software was: 35 min at 50 ℃, 5 min for the temperature increasing to 100 ℃, 22 min at 100 ℃, 5 min for the temperature increasing to 150 ℃, 24 min at 150 ℃, 5 min for the temperature increasing to 200 ℃, 25 min at 200 ℃, and then 15 min for the temperature decreasing to 50 ℃. The complete temperature cycle was about 136 min. The TD was placed upstream of the AMS. The aerosol flow can go through the heating and denuder sections (TD path) before being sampled by the AMS or flowing directly (ByPass path) into the AMS. The residence time of aerosols in the heating section was approximately 27.9 s with a flow rate of 0.45 L/min. The AMS was set with 4 menus: ByPass path in V-mode, TD path in V-mode, TD path in W-mode and ByPass path in W-mode, with 2 min in each menu. Only the data sampled during the stable temperature plateau (1839 points for V mode and 1842 points for W mode in TD-path, respectively) were selected for the calculation of volatility.

The data analysis was performed with SQUIRREL 1.57 and PIKA 1.16 (http://cires1.colorado.edu/jimenez-group/ToFAMSResources/ToFSoftware/index.html) with the method in DeCarlo et al. (2006). The mass concentration was corrected with composition-dependent collection efficiency (Middlebrook et al., 2012). All the elemental ratios calculated here were based on the Improved–Ambient (I-A) method (Canagaratna et al., 2015), while the previous Aiken–Ambient (A-A) method was also used for comparison in Table S1 in the supporting information (Aiken et al., 2008).

An aethalometer (AE-31, Magee, US) coupled with a $PM_{2.5}$ cyclone was used to measure the mass concentration of black carbon (BC) with a time resolution of 5 min. The wavelength of 880 nm was used to calculate the BC mass concentration in the data processing. The total BC ($BC_{total}$) measured by the aethalometer can be separated into BC emitted by traffic ($BC_{tr}$) and biomass burning ($BC_{bb}$) based on the aerosol absorption as described in the supplement (Sandradewi et al., 2008). A scanning mobility particle sizer (SMPS, TSI Inc.) was used to measure the particle number size distribution (mobility diameter: 15–600 nm) with a time resolution of 5 min. By assuming the densities of the components obtained in the literature (Kuwata et al., 2012; Poulain et al., 2014; Hu et al., 2017), the corresponding mass concentration can be calculated from the particle number size distribution. The AE-31 and SMPS were used only for ambient sampling.

**2.3 Particle Loss Correction**

The particle losses through the TD should be of concern for quantitative measurements with the TD, as it can decrease the transmission efficiency through the TD. There are three mechanisms of particle loss inside the TD: sedimentation,
thermophoretic and diffusional processes (Burtscher et al., 2001; Wehner et al., 2002). Burtscher et al. (2001) determined that the dominant effects would be determined by the temperature and particle size: sedimentation increases as the particle size increases and would be negligible when the TD is vertical; diffusive losses increase with decreasing particle size; and thermophoresis is not strongly dependent on particle size, is important in the denuder section and will partly compensate for diffusion in the heating section.

The transmission efficiency in this research was calculated via an experiment. CsCl was chosen as the standard chemical species for the test of the transmission efficiency of TD due to its thermal stability as discussed by Huffman et al. (2008). CsCl solution was atomized and then measured by Differential Mobility Analyzer (DMA, TSI Inc.)/Condensation Particle Counter (CPC, TSI Inc.) before and after the TD. The transmission efficiency of the ByPass path was regarded as 1. The average transmission efficiency through the TD is approximately 90%, at 50, 100, 150 and 200 ℃, as shown in Figure 1, which is
similar to previous studies (Huffman et al., 2008; Xu et al., 2016a). The transmission efficiency would be applied in the calculation of the mass concentration of particles flowed through the TD.

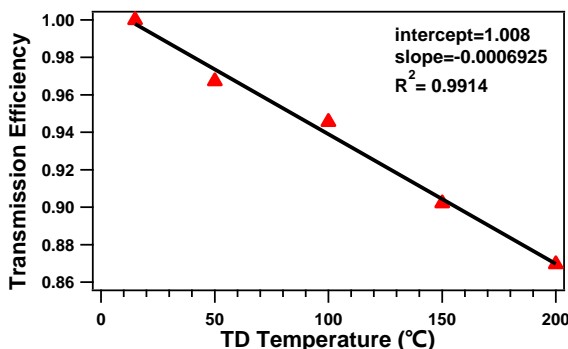

**Figure 1. Temperature related transmission efficiency through the TD.**

**2.4 Source Apportionment Method**

PMF was applied to the high resolution organic mass spectra using the PMF evaluation tool developed by Ulbrich et al. (2009). The data and error matrix were processed according to the signal-to-noise ratio (SNR) as reported in previous papers (Ulbrich et al., 2009; Huang et al., 2010; He et al., 2011). Weak ions (0.2<SNR<2) were downweighted by a factor of 2, bad ions (SNR<0.2) were removed from the analysis (Paatero and Hopke, 2003), and $CO_2^+$ related ions ($O^+$, $HO^+$, $H_2O^+$ and $CO^+$) were also downweighted (Ulbrich et al., 2009).

The PMF analysis was based on the full high-resolution organic dataset (including both data sampled under ambient temperature and that was thermally denuded) for 1 to 10 factors with fpeak varying from -1 to 1 (seed = 0), increasing with a step of 0.1, and seed varying from 0 to 250 (fpeak = 0) in steps of 10. The diagnostic plots of the solutions are shown in Figure S1 in the supplement, including the $Q/Q_{expected}$ ratio, the characteristics of the different mass spectra, the scaled residuals, and the correlation of the component time series with the external tracers. The solutions with more than five factors showed no

distinct information but splitting of the factors; the $Q/Q_{expected}$ showed the lowest value at fpeak=0; the varied fpeak did not improve the results; and the varied value of seed also made no significant difference of the solution. Therefore, the solution of five factors, fpeak = 0 and seed = 0, was determined as the optimal solution for this experiment, and the five factors are hydrocarbon-like organic aerosol (HOA), cooking organic aerosol (COA), biomass burning organic aerosol (BBOA), less-oxidized oxygenated organic aerosol (LO-OOA) and more-oxidized oxygenated organic aerosol (MO-OOA).

In addition, the PMF results with the data obtained only under ambient temperatures were also explored and the best solution was presented in Figure S2 in the supplement. Compared to the results including the thermally denuded data, the HOA and OOA were mixed to some extent, with a signature of the high fraction of $CO_2^+$ in the HOA mass spectrum. Therefore, the PMF solution with the inclusion of the thermally denuded data was confirmed as the final results for later discussion. Huffman et al. (2009b) also suggested that the PMF solution of all data collected both with and without TD-processing could facilitate the

separation of different OA factors by enhancing the contrast of the time series of these factors.

## 3 Results and discussion

### 3.1 PM$_1$ chemical compositions

Figure 2 shows the chemical compositions for only bypass conditions. Figure 2a shows the time series of the mass concentration of non-refractory species measured by the AMS and black carbon measured with an aethalometer during the

experiment. Sulfate showed a relatively stable time series, with a relative standard deviation (RSD) of 38.8%, compared to the other species, such as organics (RSD=56.1%), nitrate (RSD=69.6%), and black carbon (RSD=70.2%), indicating that sulfate was less affected by local emission sources. However, all the species decreased their concentrations largely during January 12–13 due to a heavy rain event. The sum of the non-refractory species and BC was regarded as PM$_1$, which showed a high correlation ($R^2$=0.94, slope=1.1, in Figure S3) with the mass concentration derived from the particle number concentration

measured by SMPS. In result, the average mass concentration of PM$_1$ was $42.7 \pm 20.1$ μg m$^{-3}$, ranging from 3.9 to 134.1 μg m$^{-3}$, while organics were the most abundant PM$_1$ component, contributing 43.2% to the total PM$_1$ mass concentration, followed by sulfate (21.2%), black carbon (12.2%), nitrate (11.4%), ammonium (10.4%) and chloride (1.6%). The measured and predicted ammonium showed a high correlation ($R^2$=0.97) with a slope of 0.85, implying that the aerosols showed some acidity (Zhang et al., 2007b).

The diurnal variation of the PM$_1$ species are shown in Figure 2b. The two peaks in the diurnal variation of BC obviously match the traffic rush hours at approximately 8:00 in the morning and the activities of heavy duty vehicles in the evening. When BC source apportionment was applied for our BC dataset, the results indicated that biomass burning-emitted BC also made a small contribution to the evening peak of BC (Figure S4). Nitrate showed a significant peak about 2 hours after the morning peak of BC, which was likely a result of photochemical oxidization of NOx emitted from the morning traffic. Then, the concentration

of nitrate decreased because of both the lifting of the planetary boundary layer (PBL) and its evaporation at higher ambient temperatures (also shown in Figure 2b). Nitrate maintained at a stable concentration level in the evening. Similar to ammonium nitrate, ammonium chloride is also quite semi-volatile as revealed in section 3.3. Therefore, its diurnal variation was largely influenced by the ambient temperature, as well as the height of the PBL. Also according to section 3.3, sulfate is a less-volatile species and thus would not lose significant particulate mass when the ambient temperature increases. As a secondary species

from SO$_2$ oxidation, sulfate showed a slight diurnal variation, indicating that it was less affected by the variation of the PBL. This implies that sulfate was not a typical ground-emitted species and could be better mixed in the PBL. Actually, aerosol sulfate in Shenzhen has been proved to be a species mostly from regional air mass transport (He et al., 2011; Huang et al., 2014). Since ammonium exists mostly in the forms of (NH$_4$)$_2$SO$_4$, NH$_4$NO$_3$ and NH$_4$Cl, its diurnal variation should be significantly affected by the formation processes of all these inorganic salts, besides atmospheric physical processes and semi-

volatility. The measured and predicted ammonium showed a similar correlation ($R^2$=0.96–0.97) with a similar slope of 0.84–0.85 for both the ambient temperatures and 50 ℃ (Figure S5), implying that the aerosols showed some acidity in the real ambient temperature range (Zhang et al., 2007b).The diurnal variation of organics showed more fluctuation and a few peaks, consistent with its complex origins, e.g., vehicles, biomass burning, and secondary formation (He et al., 2011; Elser et al., 2016), which will be discussed in detail in section 3.2.

The average values of O/C and H/C of organic aerosol were 0.52 and 1.61, respectively. The average O/C value in this campaign is within the typical O/C range of 0.28–0.56 previously observed under polluted urban environments in China (Huang et al., 2011, 2012; He et al., 2011; Xu et al., 2016b; Hu et al., 2016; Lee et al., 2013). The diurnal variation of O/C plotted in Figure 2c shows elevated values during the daytime, which is a clear indicator of the formation of secondary organic aerosol with more oxygen, while H/C reasonably showed an opposite diurnal trend, with decreased values during the daytime.

The quick elevation of H/C in the evening should be a combined result of various primary emissions, e.g., traffic, cooking, and biomass burning, which is supported by the source apportionment results discussed in section 3.2.

Figure 2d shows the average size distribution of the five non-refractory species. The peaks of all the species were at approximately 500 – 700 nm in the accumulation modes, while organics apparently had more mass distribution at smaller sizes down to ~100 nm. The inorganic aerosol species showed a similar average size distribution during the experiment as described

in previous studies in Shenzhen (He et al., 2011). Compared to other species, the peak of organics was slightly smaller, which was a result of the much broader size distribution of organics towards smaller sizes. This character of organic size distribution

implies that urban fresh primary emissions contributed significantly to organic aerosol (Canagaratna et al., 2004; He et al., 2011). The peak of sulfate was slightly larger than the other species, suggesting that sulfate was mostly associated with more aged particles that had grown during air mass transport (Zhang et al., 2005; Huang et al., 2008).

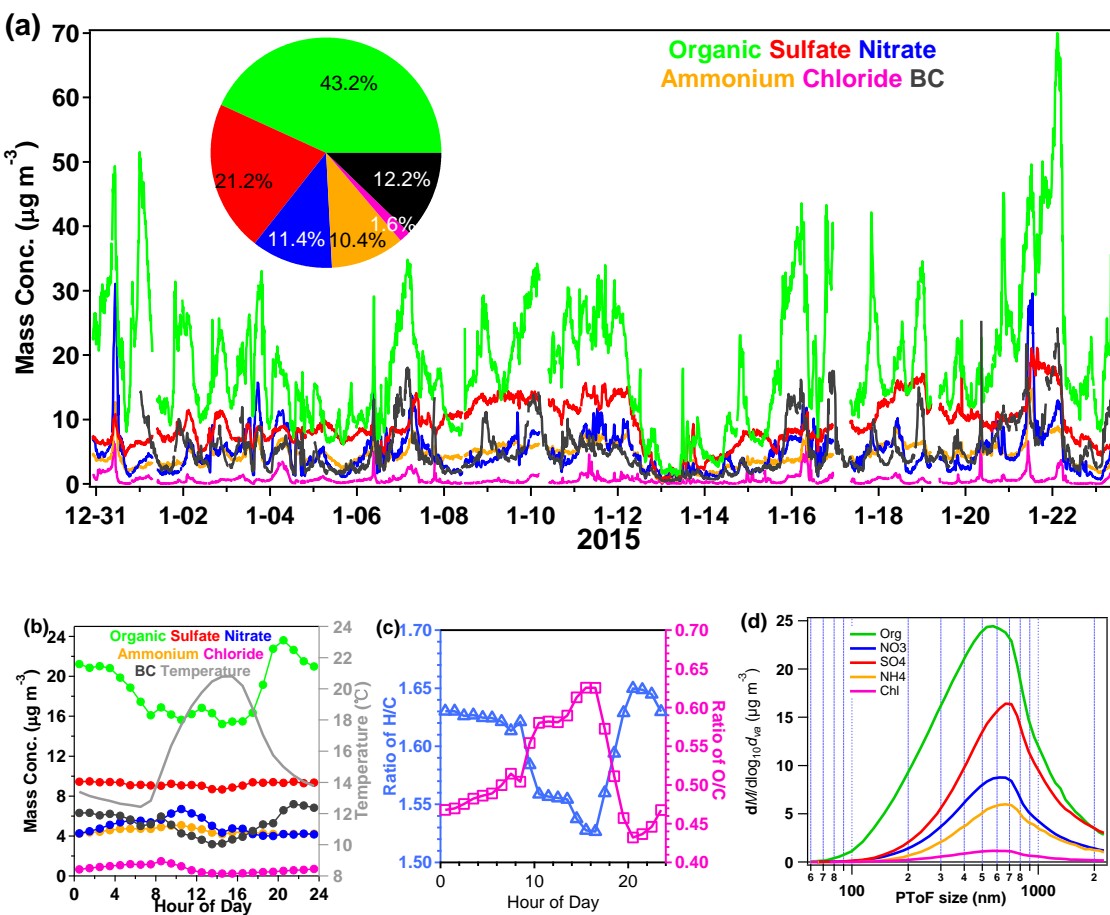


**Figure 2. Chemical composition for only bypass conditions: (a) Time series and the mass percentages of PM₁ composition; (b) diurnal variation of PM₁ species and ambient temperature; (c) diurnal variation of H/C ratio and O/C ratio; (d) average size distribution of non-refractory PM₁ species.**

**3.2 Source Apportionment**

Organics are one of the most important chemical species in aerosol pollution in Shenzhen, contributing 43.2% to the total PM₁ mass loading. As discussed in section 2.4, PMF modelling was applied to the high-resolution mass spectra of organics and five factors were identified with their MS profiles shown in Figure 3a. Under ambient temperatures, HOA, COA, BBOA, LO-OOA,

and MO-OOA averagely accounted for 13.5%, 20.6%, 8.9%, 39.1%, and 17.9% of the total organic mass, respectively (Figure 3d).

HOA is most often dominated by long chain hydrocarbon ion series of $C_nH_{2n+1}^+$ and $C_nH_{2n-1}^+$ in previous findings (Canagaratna et al., 2004; Mohr et al., 2009; Ng et al., 2010), which is also the case in this campaign. The average O/C of HOA was 0.10 in this campaign, which was in its range (0.03 to 0.17) reported in previous publications (e.g., Aiken et al., 2009; Huang et al., 2010; Mohr et al., 2012). BC is regarded as a tracer of HOA, and can be significantly emitted from both fossil fuel combustion and biomass burning (Zhang et al., 2007a; Lanz et al., 2007; Lan et al., 2011). The good correlation ($R^2$=0.82) of HOA and $BC_{tr}$ (Figure 3b) suggested that HOA was mainly from traffic emissions. The diurnal variation of HOA was influenced by PBL dynamics and also showed peaks that matched the rush hours, further supporting the dominant role of traffic emissions to the HOA.

The O/C ratio of COA is 0.18, which is similar to the previous results shown in Mohr et al. (2012). The mass spectral signature of COA is dominated by the ion series of $C_nH_{2n+1}^+$ and $C_mH_{2m+1}CO^+$ (m/z 29, 43,57,71, 85…), $C_nH_{2n-1}^+$ and $C_mH_{2m-1}CO^+$ (m/z 41, 55, 69, 83…), which are mainly ionized from alkanes, alkenes and, possibly, long chain fatty acids (He et al., 2010; Huang et al., 2010; Mohr et al., 2009; Mohr et al., 2012). Mohr et al. (2012) also identified COA from HOA by comparing the signals of m/z 55 and m/z 57 and determined that the differentiation between COA and HOA is mainly driven by the oxygen-containing ions of $C_3H_3O^+$ and $C_3H_5O^+$; especially if the signal ratio of m/z 55 to m/z 57 is larger than 2, it probably can be recognized as COA. In this study, COA showed much more $C_3H_3O^+$ than HOA, and the ratio of m/z 55 to m/z 57 showed values larger than 2, indicating the origin of cooking emissions. The diurnal variation of COA in Figure 3c shows a small peak at approximately 8:00 am, breakfast time, and the second peak at approximately 14:00 corresponds with lunch time. The mass concentration of COA also rises after 17:00, which is the time for dinner and partly because of the decreasing PBL height. The good correlation of COA with the tracer ion $C_6H_{10}O$ ($R^2$=0.91) (Sun et al., 2011; Crippa et al., 2013; Elser et al., 2016) also demonstrated the presence of COA.

The most abundant signals in BBOA are m/z 29 ($CHO^+$) and m/z 43 ($C_2H_3O^+$), and there are more fragments in the range of m/z > 100 for BBOA than for COA and HOA (He et al., 2010). BBOA can be identified by the contribution of m/z 60 (mostly $C_2H_4O_2^+$), which is distinct fragment of ionized levoglucosan, the molecular marker of biomass burning (Alfarra et al., 2007; Aiken et al., 2009; Mohr et al., 2009). Some previous studies have shown that the background level of m/z 60 /OA is approximately 0.3% in urban areas without biomass burning impacts (DeCarlo et al., 2008; Docherty et al., 2008). The signal of m/z 60 in BBOA is 1.36% in this study, indicating the presence of BBOA during this experiment, and the BBOA correlated well with m/z 60 ($R^2$=0.83). The O/C ratios of BBOA varied a lot in previous studies. Laboratory studies reported O/C ratios of 0.18–0.26 for six types of biomass burning emissions (He et al., 2010), and O/C ratios of 0.31 for lodgepole pine burning and 0.42 for sage/rabbitbrush burning (Aiken et al., 2008). DeCarlo et al. (2010) reported an O/C ratio of 0.42 for ambient

biomass burning aerosol. The BBOA in this study showed an O/C ratio of 0.33, which is within the range of previous studies. The diurnal trend of BBOA showed a large peak in the evening, well consistent with the diurnal peak of $BC_{bb}$ in Figure S4.

OOA is recognized by the most intense signal of m/z 44 ($CO_2^+$), and the signals at higher values of m/z are lower relative to those of other OA factors (Ng et al., 2010). Furthermore, the OOA can be divided into two factors, LO-OOA (typically named semi-volatile OOA) and MO-OOA (typically named less-volatile OOA), according to the O/C ratios and f44 (Jimenez et al.,
2009; Ng et al., 2010; Xu et al., 2015). The factor with a relatively higher O/C ratio (0.95) of OOA and higher f44 than f43 is identified as MO-OOA. It showed a good correlation ($R^2$=0.64) with sulfate, which was less volatile and had been identified as a regional pollutant in Shenzhen (He et al., 2011; Huang et al., 2014), implying MO-OOA could also be aged aerosol from regional transport. The correlation of MO-OOA with nitrate ($R^2$=0.27) is much lower than with sulfate, which is a typical result. Meanwhile, LO-OOA, which is less oxygenated (O/C = 0.76), showed a narrow gap between f43 and f44. The
correlation of LO-OOA with sulfate ($R^2$=0.59) was indeed a little bigger than with nitrate ($R^2$=0.46), which could be a combined result of the precursors, formation mechanisms, as well as volatility of LO-OOA. Unlike the primary organic components, which had lower concentrations during the daytime due to the elevated PBL, the diurnal variations of both LO-OOA and MO-OOA showed higher concentrations during the daytime, suggesting that photochemical secondary production should be their main source. The diurnal variation of MO-OOA was relatively stable compared to that of LO-OOA, which is
consistent with that MO-OOA is a more aged and regional component. The contribution of the two OOAs to total OAs is 57.0%, indicating that OOA contributes the majority of the OA pollution in wintertime in Shenzhen.

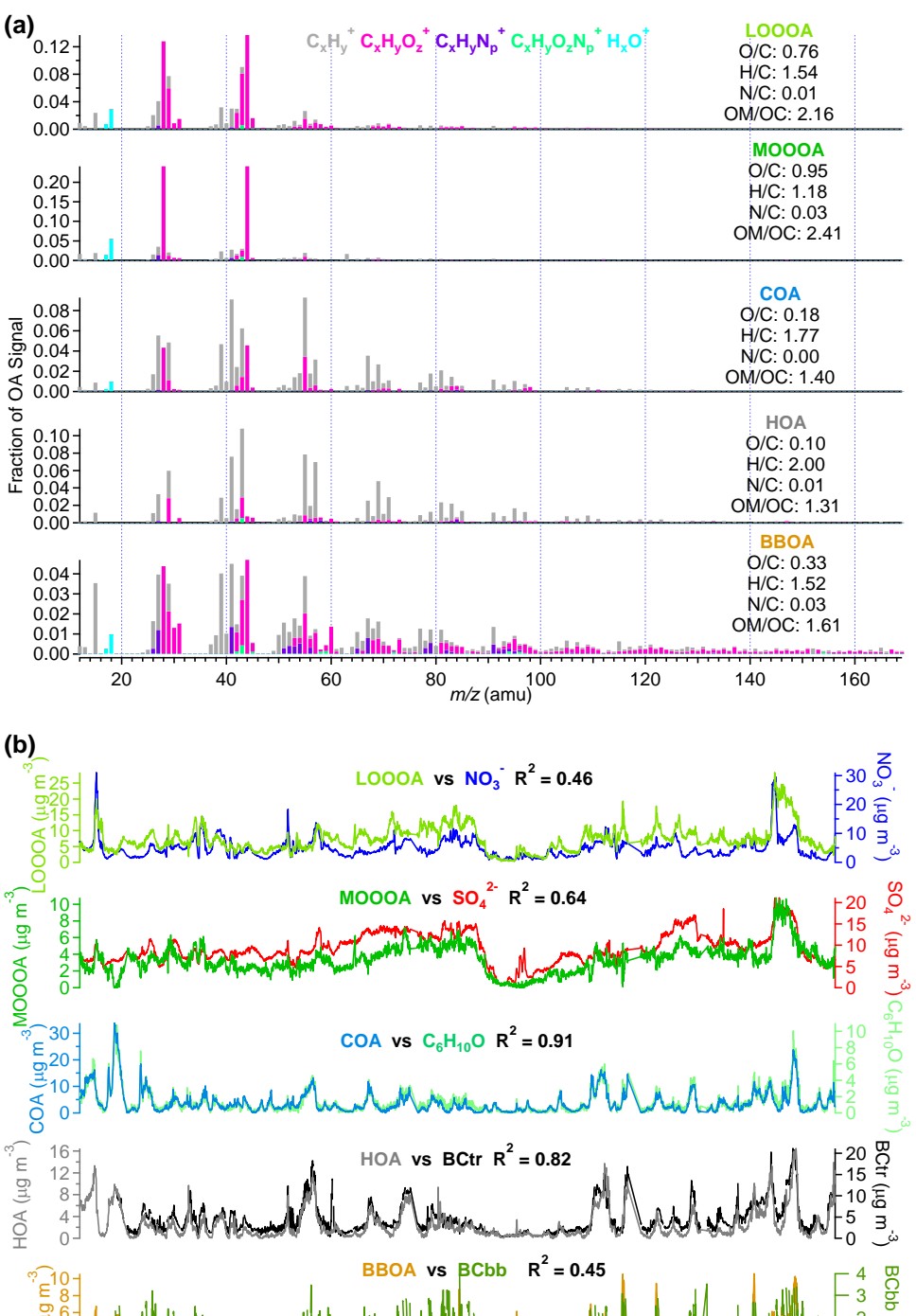

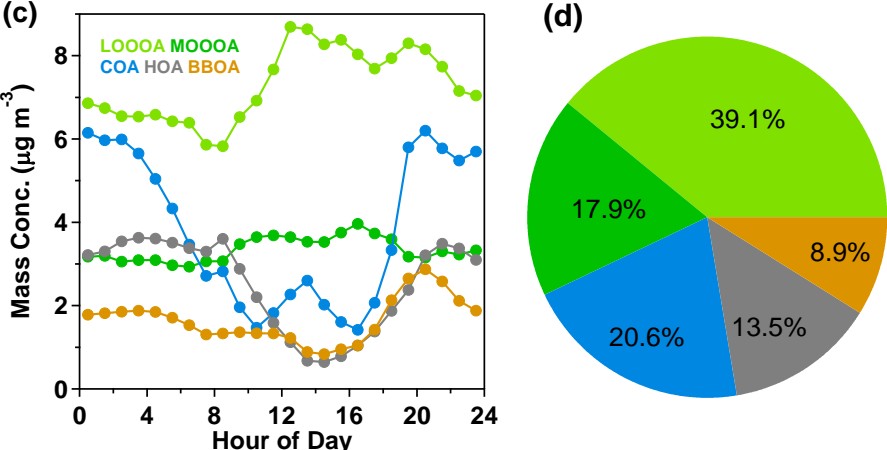

**Figure 3. (a) MS profiles of the five OA factors identified by PMF; (b) time series of the five OA factors and the correlation with the relevant species during the experiment; (c) the diurnal variation of the five OA factors; (d) the average contributions of the five OA factors to total OAs. The time series, diurnal variation and pie chart display the results only for ambient conditions.**

Figure 4 showed the mass fractions of the five factors at different TD temperatures. It is found that when the temperature increasing, the fraction of MO-OOA quickly increased up to 67.6% at 200 ℃, while LO-OOA showed a reverse trend, accounting for only 2.9% at 200 ℃, indicating that they had quite different volatilities. For HOA, COA, and BBOA, they also exhibited different volatilities, with HOA accounting for only 5-7% above 100 ℃, while the fraction of COA did not change much with the temperature increasing. The different volatilities of different OA factors will be discussed in more detail in the following section.

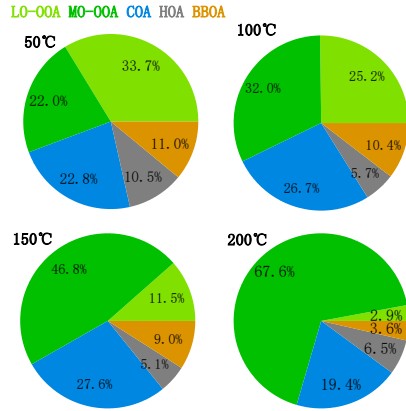

**Figure 4. The average compositions of the total OA at different TD temperatures.**

### 3.3 Volatility of PM$_1$ species and OA factors

Figure 5 shows the mass fraction remaining (MFR) of the non-refractory species. The MFR is calculated as the ratio of the species mass concentrations with and without TD-processing. The narrow average MFR $\pm$ standard deviation regions show that the volatilities of these species were stable during the measurement. The MFRs of the total non-refractory species and organics both showed nearly linear correlations with the TD temperature, which is consistent with the fact that they include various compounds with a wide range of volatilities. For organics, the MFR was 0.88 at 50 ℃, 0.63 at 100 ℃, 0.32 at 150 ℃ and, finally, 0.16 at 200 ℃. Based on the linear relationship of the MFR of organics and the TD temperature, the evaporation rate of organics is approximately 0.45% ℃$^{-1}$. The relationship of the O/C, H/C, and N/C ratios with the TD temperature is also shown in Figure 5. It can be seen that O/C kept increasing as the temperature increased, especially after 150 ℃, which is consistent with previous studies (Xu et al., 2016a). When examining the organic mass spectra at difference temperatures (in Figure S6), the elevation of O/C with temperature increasing was found to be reasonably related to increasing of $CO_2^+$. Previous PMF results usually correlated higher volatility with reduced species and lower volatility with more oxygenated species (Ng et al., 2010; Huang et al., 2012). In this study, the elevation of O/C with temperature increasing was closely related to the evaporation of more reduced organic components, as the PMF results indicated. On the other hand, H/C showed a reasonable reverse trend relative to that of O/C. N/C generally had an increasing trend with the TD temperature increasing, but N/C varied largely at the different TD temperatures, suggesting that the volatilities of N-containing compounds are complex.

Figure 5 also shows the MFRs of inorganic species with different temperature, and all the inorganic species showed trends similar to the results shown by Huffman et al. (2008). Nitrate and chloride show similar trends: they both decreased sharply from ambient temperature to 50 ℃, reaching approximately 0.57; then decreased at a lower rate from 50 to 150 ℃. When the TD temperature increased to 200 ℃, the MFRs of nitrate and chloride were at approximately 0.08, which are lower than those of all the other species. Sulfate is the least volatile species among all PM$_1$ species. The MFR of sulfate does not show a significant decreasing trend with temperature below 100 ℃ (0.93 at 50 ℃ and 0.89 at 100 ℃); when the temperature reached 150 ℃, the MFR decreased sharply to 0.43, with 11% left at 200 ℃. The trend of MFR of sulfate is consistent with the discussion in Burtscher et al. (2001): sulfuric acid would evaporate under temperatures of 30–125 ℃, while ammonium sulfate and bisulfate would evaporate between 125 to 175 ℃.

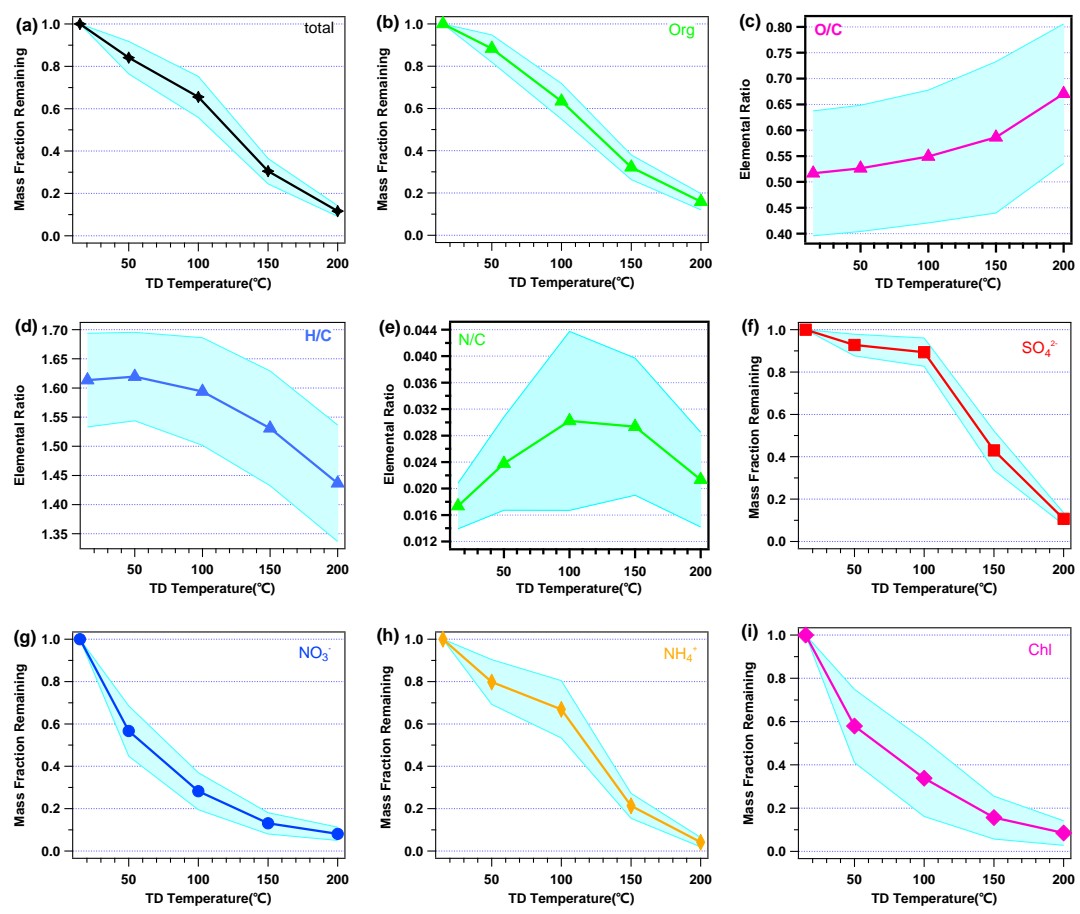

**Figure 5. Variation of the average mass fraction remaining (MFR) of the total of non-refractory species and the ratios of O/C, H/C, and N/C with the TD temperature. The shaded regions indicate the average ± standard deviation.**

Figure 6 shows the MFRs of the different OA factors identified by PMF. The MFR of HOA was 0.56 at 50 ℃ and decreased by 1.26 % ·℃⁻¹ from the ambient temperature to 50 ℃, and got to only 17.8% at 100 ℃. Then, the evaporation rate slowed down significantly, with a MFR of 7.6% at 200 ℃. BBOA showed significant volatility in this study, with an evaporation rate of 0.37% ·℃⁻¹ near the ambient temperature and an MFR of 0.87 at 50 ℃. In addition, it was noted that the evaporation rate of BBOA got quicker when the TD temperature increased from 100 ℃ to 150 ℃, indicating that BBOA contained more compounds that favour evaporating in this temperature range. COA showed a similar volatility to that of BBOA, with an MFR of 0.85 at 50 ℃, but it kept a stable evaporation rate throughout the whole temperature range (~0.44% ·℃⁻¹). The volatility of LO-OOA (MFR=0.70 at 50 ℃) was higher than those of MO-OOA (MFR=0.99 at 50 ℃), COA and BBOA, but lower than that of HOA. The MFR curve of MO-OOA remained relatively stable below 100 ℃ and then decreased significantly when the temperature increased further. The MFR of MO-OOA was as high as 51.5% even at 200 ℃. Thus, MO-OOA was the least volatile one among the five factors. The OOA (LO-OOA+MO-OOA) in Figure 6 presents the MFR

variation of the combination of LO-OOA and MO-OOA. The MFR of OOA, which is regarded as a good surrogate of SOA, showed a good linear decline at the temperatures under 150 ℃, with an evaporation rate of 0.54 % ·℃⁻¹. The evaporation rates of the different OA factors identified by PMF in this study provide the first hand information of semi-volatility of
organic aerosols in China, which will be helpful in constraining parameters in the atmospheric chemical models.

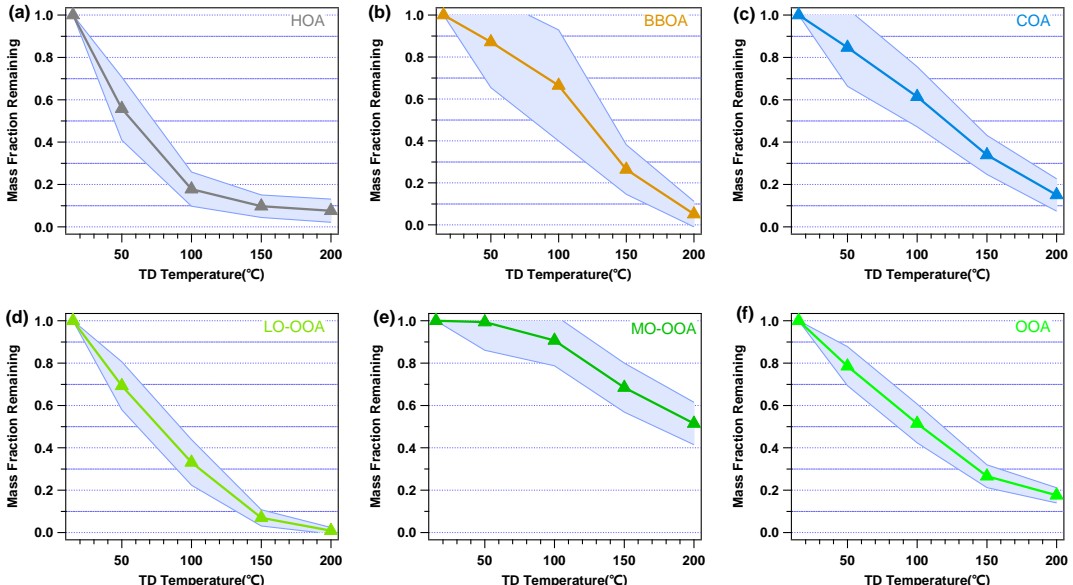

**Figure 6. Average mass fraction remaining (MFR) of the five OA factors and OOA (calculated as the combination of MO-OOA and LO-OOA). The shaded region is the average MFR±standard deviation.**

Figure 7 compares both the volatilities (MFR at 50 ℃) and the O/C ratios of the five factors. The sequence of the volatilities can be summarized as HOA > LO-OOA > COA ≈ BBOA > MO-OOA. It can be easily found that the sequence of the volatilities of the OA factors does not completely follow the sequence of the O/C ratios. For example, although LO-OOA has a higher O/C ratio than BBOA and COA, LO-OOA is also more volatile (or with a lower MFR) than BBOA and COA. This clearly indicates the volatility of the OA factors depends not only on the oxygenation of organic compounds, but also other
factors, e.g., molecular weight and mixing state. HOA is identified as the most volatile OA factor while MO-OOA is nearly non-volatile near the real atmospheric temperatures in Shenzhen, which is consistent with the results observed in Mexico and Paris (Cappa and Jimenez, 2010; Paciga et al., 2016). However, LO-OOA is the second volatile OA factor after HOA in Shenzhen, which is different from that in Mexico, where BBOA is more volatile than LO-OOA. Actually, the volatility of the aerosols directly from biomass burning have been measured to be quite variable, with an evaporation rate of 0.2–1.6% ·℃⁻
¹, depending on the kinds of wood and combustion conditions (Huffman et al., 2009a). The relatively lower volatility of COA was also identified in previous studies and attributed to the abundant fatty acids of low volatility in COA (Mohr et al., 2009; Paciga et al., 2016). Hong et al. (2017) recently reported the estimation of the organic aerosol volatility in a boreal forest in

Finland using two independent methods, including using a VTDMA with a kinetic evaporation model and applying PMF to HR-AMS data. Semi-volatile and low-volatility organic mass fractions were determined by both methods, similar to our study in China. This implies that MO-OOA and LO-OOA, with different volatilities, could be popular organic aerosol components across the world. Hong et al. (2017) also pointed out that determining of extremely low volatility organic aerosols from AMS data using the PMF analysis should be explored in future studies.

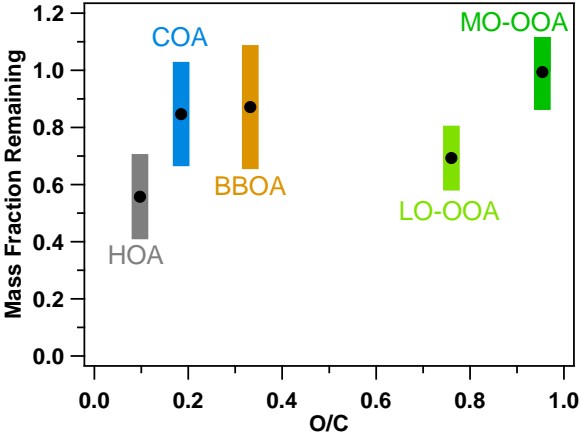

**Figure 7. Comparison of the volatilities (MFRs at 50 °C) and O/C ratios of the five OA factors in this study. The colourful regions indicate the standard deviations of the MFRs.**

It should also be noted that, as the most volatile species in this study, HOA could be evaporated easily and thus have a larger potential to experience the "evaporation–oxidation in gas phase–condensation" process and then form SOA, since previous studies showed that semi-volatile hydrocarbons from diesel exhaust (Robinson et al., 2007) and crude oil (de Gouw et al., 2011) can be easily oxidized in the gas phase to form less volatile SOA. Huang et al. (2012) also pointed out that HOA could be oxidized to SOA based on the analysis of their diurnal variations. This potential can be further supported by the fact that it was difficult to resolve HOA in downwind regions far from urban and industrial areas in China, implying that HOA had been oxidized during the air mass transport (Huang et al., 2011; Zhu et al., 2016). Therefore, the modelling work needs to consider the process from HOA to SOA in future.

## 4. Conclusions

The source apportionment and volatility of the $PM_1$ chemical composition during winter in Shenzhen were investigated based on the TD-AMS system. The mean $PM_1$ mass concentration was $42.7 \pm 20.1$ μg m$^{-3}$ during the experiment, with organics (accounting for 43.2%) as the most abundant species. Sulfate, black carbon, nitrate, ammonium and chloride contributed 21.2%, 12.2%, 11.4%, 10.4% and 1.6% to the total $PM_1$, respectively. The chemical species in $PM_1$ exhibited a range of

volatilities. Nitrate showed the highest volatility among the five species measured with the lowest MFR (0.57) at 50 ℃.

Organics exhibited a relatively linear MFR decrease, with a rate of 0.45 % ·℃$^{-1}$, as the TD temperature increased from ambient to 200 ℃, which is mainly due to the complex composition of organics in the atmosphere. The organics were grouped into five subtypes by the PMF analysis, including primarily emitted HOA, COA, BBOA and two secondarily formed oxygenated OAs: LO-OOA and MO-OOA, and they accounted for 13.5%, 20.6%, 8.9%, 39.1% and 17.9% of the total OA, respectively. Among all the five OA factors, HOA was the most volatile species, whereas MO-OOA had the lowest volatility with the

slowest evaporation rate. According to the MFR of different OA factors at 50 ℃, the volatility sequence of the five OA factors was HOA (MFR of 0.56 at 50 ℃) > LO-OOA (0.70) > COA (0.85) ≈ BBOA (0.87) > MO-OOA (0.99), which was not completely consistent with the sequence of their O/C ratios. The most volatile HOA had a high potential to be oxidized to secondary species in the gas phase.

**Acknowledgments**

This work was supported by the National Natural Science Foundation of China (U1301234, 41622304), and the Science and Technology Plan of Shenzhen Municipality (JCYJ20170412150626172, JCYJ20170306164713148). We thank Prof. André S. H. Prévôt of Paul Scherrer Institute, Switzerland, for providing the code for BC source apportionment.

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
