# Peer review of "Volatility measurement of atmospheric submicron aerosols in an urban atmosphere in southern China"

_Atmospheric Chemistry and Physics, 2017_

## Referee Comment (RC1) · Anonymous Referee #1 · 15 Sep 2017

This manuscript by Cao et al. presents the field measurement results using an AMS coupled with a TD conducted in China. As the application of TD-AMS is for now still rare in the atmospheric chemistry studies in China. The manuscript contains useful and valuable information. However, this reviewer finds that in the current version, many necessary details are missing, and many statements and important conclusions appear to be over-interpreted, not well supported by the data or at least is not well explained, which limits the scientific value of the work. The current version needs major revision before its acceptance. The current version fails to make good use of the wealthy data obtained by the HR-AMS, and it should be extensively expanded. Here below I list the comments below that need to be addressed.

(1) Please pay attentions to the in-text citation format
(2) In the introduction part, please provide more details regarding previous TD-AMS studies, focused the important findings.
(3) Introduction: Recently, Xu et al. (Atmos. Chem. Phys. 2014, 14, 12593-12611), and Li et al. (Atmos. Environ. 2017, 158, 270-304.) have provided a summary of the AMS studies in China, these two papers should be cited.
(4) Section 2.2: The authors tested the transmission efficiencies, do these numbers are included to calculate the MFR? Besides, some theoretical calculations by Chris Cappa et al. regarding TD kinetics should be included.
(5) Section 2.2: why not to try a higher TD temperature, for example 250C or even 300C, there can be extremely low volatility species remained at such temperatures
(6) Section 2: Some details regarding the TD is not provided. For example, what is the temperature cycle. What is the sampling time at each TD temperature? Do the authors change the absorption materials of the TD? And how often?
(7) Section 2: A lot of details are not provided. You should give a bit more details regarding the instrumental and operational details about the AMS, for the convenience of readers not in the AMS community. Calibrations? Relative ionization efficiencies? Size calibration? What are the chemical resolution? As your PMF includes ions with m/z up to 170, does these ions can be well separated from their neighborhood ions? Any instrument details about AE31?
(8) Section 2.3: As this study are the very first few ones that used combined TD and Bypass data to perform the PMF analyses, more details should be provided. The determination of optimal solution, what are the profiles of other solutions? What are the diagnostic plots of the currently chosen solution? What are the correlations of the factors with external tracers? what the PMF results are if only consider bypass data, and then to show clearly what are the advantages (please provide such information in the forms of plots in details not only a brief description) by running PMF on the combined TD and Bypass data. Such details are missing, but they are very helpful and also necessary to evaluate the robustness of your PMF results.
(9) Results: P5L20 It is not convincing to state that sulfate is due to regional transport while other species are influenced by local emissions, only based on the extents of data variability (and the variability can be quantified, from xx to xx and what are the standard deviations?)
(10) P5L25: the BC contribution seems to be quite high in PM1 (12.2%), is this number

consistent with previous values in Shenzhen? Is this due to heavy traffic?That's also why I asked the calibration and operational details of AE31.

(11) P5L25: Is this composition for bypass conditions or others? Please indicate it clearly.

(12) P5L29-: Please add the meteorological parameters, such as temperatures, solar radiation data to help explain the variations of nitrate. For example, it is not convincing to state the photochemical oxidation of NOx only based on the peak after BC peak. Similar for chloride, NH4NO3 is also semi-volatile, so why G/P partitioning drives chloride but not nitrate? Also, if you have PBL data, then add its diurnal pattern as well.

(13) P6L7: Why not provide the neutralization plot? You can do that to support your statement.

(14) P6L12: Do the elemental ratios calculated by Aiken 2008 method, or Canagaratna 2015 method? Please explain. Also, if it is from Canagaratna 2015 method, can it compare with previous values fairly?

(15) P6L23: I don't think "which may be caused by the internal mixing and similar gas-particle distribution processes" is correct. If this is correct, then why you state previously that nitrate is due to photochemistry while chloride is due to G/P partitioning? They are not consistent.

(16) P6L25: "products of photochemical reactions of VOCs have a significant influence on the organic pollution" I don't think your data presented till here can indicate this. And why only photochemical reactions of VOCs, and other possible processes are excluded?

(17) P6L27: why a large size would mean a more aged sulfate? and why it would then mean it had been transported regionally?

(18) P7L6: Is your AE31 downstream of the TD? If so, can you plot the MFR of BC measured by AE31? You didn't mention the instrument setup of AE31 at all.

(19) P8L7: "sulphuric acid can be distinguished with temperature between 30 and 125 °C", what does this mean? I don't quite understand.

(20) P8L10: this paragraph is not well written. You should provide the average OA MS at different TD temperatures to give more explanations. You didn't explain why a more rapid increase occurred from 150-200C, and "OA is heated and oxidized into more aged compounds", there is lacking any supporting evidence. Jimenez et al., 2009 does not say so. "the results is consistent with the OA evolution process (Jimenez et al., 2009)?" Please be more specific and not so vague, this paper doesn't have any TD-AMS results, although it does show that the O/C ratio of OA increases with the decrease of saturation of OA.

(21) P9L15: In my viewpoint, your BC and HOA diurnal patterns don't have clear matched with traffic times. It is overall high during evening to noon, and low during afternoon, more relevant to PBL dynamics, likely.

(22) P9L23: Your writing logistic is strange. Sometimes you mentioned the results of previous studies, normally, the readers expect to see what your data show, but instead you didn't mention them. Here, you mention the findings of Mohr et al., 2012 regarding mz. 55 and 57, then how does them behave in your MS? If you are not going to describe them for your data, then why you mention Mohr's results? Their results don't support your explanation.

(23) P9L28: That's what I am talking about, you mention PBL influence on COA, why not on HOA, etc.

(24) P10L5: Do you have data of external BB tracer to use?

(25) P10L15: What are the correlation coefficients of LOOOA with sulfate, as well as MOOOA with nitrate? Typically, the important criteria is to compare the correlations coefficients, and then postulate the assignment of LOOOA and MOOOA (although it is not always the case.) It seems like LOOOA might have a better correlation with sulfate, while MOOOA is better correlated with nitrate from Figure 4b, how to explain? What are the cross correlation coefficients between diurnal patterns of these species? A more comprehensive discussion regarding these SOA should be combined with meteorological data and also gaseous species, if available.

(26) P10L20: When you talk about the mass contributions of different OA factors, do you mean the average contributions over all data including bypass and TD? Or it is only for the case of bypass conditions? You didn't indicate this clearly at all throughout the manuscript.

(27) P12Section 3.4: Can you show the mass fractions of different OA factors at different TD temperatures?

(28) P12Section3,4: As I suggested before, this section doesn't discuss any compositional changes of OA, but only mention the variations of MFR. You have an HR-AMS which can provide chemical information in very details, which is in fact the unique advantage of HR-AMS, why not do so?

(29) P13L10: It is not that convincing to state that HOA rather than BBOA or COA was a more important source of LO-OOA, based on only the volatility sequence. The sequence may change at different temperatures, then the conclusion will change according to the logistic used here. To prove this, molecular level analyses are required. For example, a HOA tracer compound get oxidized, and the products appear in LO-OOA. Also, heterogeneous or condensed-phase reactions may also produce LO-OOA related species, it is not definitive that the reactions have to proceed via evaporation-gas-phase oxidation-condensation. Such statement is high speculative without any solid evidence in my opinion.

(30) P13: Basically, I think it is dangerous to state the evolution process of different OA factors based on only volatility data. Even you are lacking of other data, you can perform more analyses, such as the temporal variations, compositional changes, correlations with other species, and discuss these in more details. This can be done for a few cases

---

## Referee Comment (RC2) · Anonymous Referee #3 · 22 Sep 2017

In this study, the authors deployed a TD-AMS to investigate the volatility of different chemical compositions of PM1 in Shenzhen. Aerosol volatility studies are important but rare in China. This work, as far as I know, could be the 1st report on the online measurement of the volatilities of aerosol chemical components using a TD-AMS in China, and thus provide valuable information. In addition, OA was classified into several groups using PMF and tested their volatility separately, which help to understand their sources and characters. The manuscript is overall well written and documented. The topic fits well in the scope of ACP. I recommend this manuscript can be published after some revisions.

[Figure]

1) In general, aerosol is a really complex system, especially in China. The analysis in this work is a bit too simple. Some more discussions are recommended, e.g. size-revolved volatility study, organic aerosol volatility in different events, e.g. heave haze events or NPF events.

2) The last sentence of conclusion part, "HOA, rather than BBOA or COA, could be a potentially important source of LO-OOA". More discussions are needed to support this statement.

3) Hong et al., 2017 has reported a similar work that estimate of the organic aerosol volatility using two independent methods including a VTDMA and HR-AMS. They compared the direct measurement result from VTDMA and PMF result from HR-AMS. It would be good to add some discussions to compare the methods and results between this two works.

Hong, J., Äijälä, M., Häme, S. A. K., Hao, L., Duplissy, J., Heikkinen, L. M., Nie, W., Mikkilä, J., Kulmala, M., Prisle, N. L., Virtanen, A., Ehn, M., Paasonen, P., Worsnop, D. R., Riipinen, I., Petäjä, T., and Kerminen, V. M.: Estimates of the organic aerosol volatility in a boreal forest using two independent methods, Atmos. Chem. Phys., 17, 4387-4399, 10.5194/acp-17-4387-2017, 2017.

4) I suggest adding a summary of volatility studies in China. Although there should be no other studies using a TD-AMS, but some related work using VTMDA are still worth to be summarized, e.g. Cheung et al., 2016; Nie et al., 2017.

Cheung, H. H. Y., Tan, H., Xu, H., Li, F., Wu, C., Yu, J. Z., and Chan, C. K.: Measurements of non-volatile aerosols with a VTDMA and their correlations with carbonaceous aerosols in Guangzhou, China, Atmos. Chem. Phys., 16, 8431-8446, 10.5194/acp-16-8431-2016, 2016.

Nie, W., Hong, J., Häme, S. A. K., Ding, A., Li, Y., Yan, C., Hao, L., Mikkilä, J., Zheng, L., Xie, Y., Zhu, C., Xu, Z., Chi, X., Huang, X., Zhou, Y., Lin, P., Virtanen, A., Worsnop,

[Figure]

D. R., Kulmala, M., Ehn, M., Yu, J. Z., Kerminen, V. M., and Petäjä, T.: Volatility of mixed atmospheric Humic-like Substances and ammonium sulfate particles, Atmos. Chem. Phys. Discuss., 2016, 1-26, 10.5194/acp-2016-839, 2016.

---

## Referee Comment (RC3) · Anonymous Referee #2 · 29 Sep 2017

This manuscript describes the volatilities of the PM1 chemical components by using the Thermo-Denuder – Aerosol Mass Spectrometer (TD-AMS) system, along with the positive matrix factorization (PMF) analysis. The results make some very important implications on the atmospheric chemistry of aerosol particles, as there appears to be the first report about such study under a polluted environment in China. Overall, the content of this study fits within the scope of ACP. I agree that this campaign was well-designed. However, the authors need to consider making more further clarifications/evidences to support a couple of ambiguous discussion and/or conclusions in this paper. Numerous corrections on the text editing are needed, including reference and abbreviation formats as well as language issues, etc. Please the authors carefully

check that throughout the manuscript. Therefore, a major revision is needed before it would be accepted in ACP.

General comments:

When I finished reading this manuscript, I feel like that the authors did not fully analyze such comprehensive data set, then, highlight the new findings during the discussion of this study, because I found a lot of "consistent comparison" between this study and previous studies. I can understand that the authors would like to support your results/discussion/conclusions, and I am not saying that you should not do that. But the authors should try to find something new as those comparisons with previous works.

More analysis could also be done to understand such data set. For example, for both TD-path and non-TD-path data: temporal variations (PM1 species and PMF-OA factors)? chemical changes under the different environment conditions? any evidence for potential origins of changing volatilities for these species (e.g., what's difference between marine and continental air masses)? Variations of size distributions, rather than averaged ones? etc.

More details of experimental materials need to be shown wherever in the main text or supplementary. What's the duration for TD-path data? It's easier to understand for readers if the authors could show that, for instance, in the time series of Figure 2a. What's the time resolution of your measurements during the campaign? How did the authors calibrate the AMS, and what were the results, e.g., values of IE, RIENH4, RIESO4? The authors should show one figure for the relationship between measured NH4 and predicted NH4 for TD-path and non-TD-path data, respectively.

I am not convinced by the state of a finding about "...that HOA, rather than BBOA or COA, could be a potentially important source of LO-OOA...", as shown in the abstract and the main text elsewhere, just based on current TD-AMS-PMF results. The authors should perform more analysis to support that. For example, typical cases analysis?

[Figure]

Since the authors have the data of seven-wavelength light absorption, it will be useful to support your PMF-POA factors by performing source apportionment of black carbon (BC) with aethalometer data (Elser et al., 2016).

Comments/suggestions in details:

Please note that abbreviations should be used in the same format throughout all the manuscript. For example, page 1, line 12: "a TD-AMS (Thermo-Denuder – Aerosol Mass Spectrometer)" and page 1, line 20: "a hydrocarbon-like OA (HOA, . . .)". For the consistency, the authors may replace "a TD-AMS (Thermo-Denuder – Aerosol Mass Spectrometer)" by "a Thermo-Denuder – Aerosol Mass Spectrometer (TD-AMS)". Somewhere else if the same issue should also be done.

Please define abbreviations when using it for the first time. For instance, Page 1, line 13: submicron particulate matter (PM1); page 1, line 19: positive matrix factorization (PMF). Somewhere else if the same issue should also be done.

Page 2, lines 15-25, the authors should also introduce more about the major findings reported by those previous studies. Then, the authors may tell readers the missing knowledge according to the new findings of your study.

Page 2, lines 29-31, I cannot understand the relationship of this sentence with the major story of this introduction.

Page 3, line 15, Duplicate definition for "thermo-denuder (TD)", it has been defined in the first time in Page 2 line 15. In addition, abbreviation should be followed hereafter when it has been defined at the first time. The authors should carefully check the similar issues as others, e.g., black carbon, organic aerosol, etc.

Page 5, lines 9-15, it's hard to read these sentences Please re-edit. The authors may introduce your data treatment procedure, then/at the same time, you could give the reference(s) to support yours as well as explain why.

Page 5, line 23, the AE-31 should be described in the experimental method section.

More details should be also given, e.g., cutoff, and which wavelength you used for the equivalent black carbon concentration. It is same for SMPS in line 29, for which more description should be given in this section too.

Page 5, lines 24-25, "…due to rain…", to state this, the authors should provide related rain data to prove it. And what's the link of "…sulfate showed a relatively stable…" to this "rain case" in this sentence? and I don't understand why the relatively stable time series of sulfate can be considered as regional transportation? The authors may perform more analysis on chemical species along with your ground-measured meteorological parameters. Also, for instance, air mass trajectory analysis would be also useful to help figure it out.

Page 5, lines 23-30: It's hard to read such long sentence. The authors should separate it for each information what you want to discuss. Such kind of long sentences is also frequently showing somewhere in this paper. The authors should keep the similar modification.

Page 6, lines 2-3: The authors should make the plot to support this discussion. And it would be also interesting to see what's the different ratio of measured and predicted NH4 from TD-path and non-TD-path data.

Page 6, line 4: Double meanings between "diurnal variation" and "during the day" in one sentence. Please reword it and somewhere same is also needed.

Page 6, line 5: Please the authors provide any evidence to prove the contribution of "the activity of heavy duty vehicles" to BC in the evening. If it's a case, and what's the difference sources that contribute BC particles between morning and evening peaks? Indeed, I feel more like that biomass burning emissions (according to the next discussion of BBOA variations) might also contribute the evening peak of BC. That's also one of reasons that I propose the authors to perform the BC source apportionment.

Page 6, lines 6-10: The authors should be careful to sate the nitrate variations just

according to such diurnal peaks between BC and nitrate. For example, how did the authors indicate that the peak of nitrate after the BC one should be linked to photo-chemistry, and that the peak at around 14:00 is due to "the enhancement of sunlight"? And, why there was no influence of gas-particle partitioning on nitrate, as discussed only for chloride?

Page 6, lines 10-11: The authors should provide/link your evidence or any published work(s) to prove such kind of discussions. In the manuscript, somewhere else with the similar issue should be modified too.

Page 6, lines 11-13: Remove "during the day". I cannot understand that "...a regional product of oxidation by SO2 that is transported ...", since I did not see the transported evidence of sulfate in this study. The authors may further analyze the temporal variations along with the size distribution of sulfate, and considering meteorological influence.

Page 6, lines 14-15: Please provide the neutralization plot. It seems an odd sentence for "...so the diurnal variation of ammonium was influenced by sulfate, nitrate and chloride". It's generally true that ammonium measured by the aerodyne AMS is mainly in the form of ammonium sulfate, ammonium nitrate, and/or ammonium chloride. I do more trust that diurnal variations of ammonium can be also affected by such inorganic salts formation processes, besides other factors, e.g., atmospheric physical processes. So, the authors should reword it.

Page 6, lines 15-16: I don't think the authors need to repeat such information of organic aerosols as already provided in the introduction before (page 2 lines 13-14). introduced before. Again, somewhere else, such kind of discussion, at least, the authors should provide reference(s) to support it. I suggest the authors to reedit and combine this sentence with the next one (lines 16-18).

Page 6, lines 19-21: It's complicate to read here with a lot of comparison in only one sentence. Were all the averaged reference values only from Chen et al. (2015)? In

addition, why did not the authors compare those values of O/C and H/C with some results observed under other polluted environments of China? It might make sense to understand such knowledge over the regional scale for developing countries in Asia (e.g., China).

Page 6, lines 21-25: The authors did not explain those diurnal variations of O/C and H/C. And why only a small H/C peak at noon was discussed, but no explanation at the peak during the nighttime? Do the authors think the biomass burning could also influence H/C variations, in addition to traffic and cooking emissions?

Page 6, line 26: The authors should avoid highlighting "non-refractory species measured by the AMS" too many times over the manuscript, because readers will know that after you explain it at the first time (except for the special case). Please the authors carefully check that elsewhere.

Page 6, lines 26-28: Be careful making the conclusion of averaged "approximately 500 – 700 nm in the accumulation modes" linking to "all the species" being aged particles. For example, were "all the species" including primary emissions, as below discussed HOA and BBOA, as well as fresh OOA? In addition, the authors already discussed that nitrate can be formed just after the morning traffic rush hours, so is this also included? To understand so, the authors can do the time series of size distribution of each chemical species (including both inorganic aerosols and PMF-OA factors) instead of showing here.

Page 6, line 30: I don't understand why "a similar average size distribution" of these inorganic species is because of this.

Page 6, lines 30-32: As reported the comment of 21, the authors should provide the size distribution of PMF-OA factors to prove this. In addition, how to prove "products of photochemical reactions of VOCs have a significant influence on the organic pollution.", while rather than other formation processes?

[Figure]

Page 7, lines 1-2: How to understand here, the large size of sulfate being aged and from regional transports?

Page 7, line 8: Again, "measured by the AMS", such kind of words, does not need to be iterate.

Page 7, line 10: What does mean by "measured directly by the AMS."? Is it meaning the measured particles from non-TD-path channel?

Page 7, lines 12-13: The authors stated "...of the total non-refractory species and organics all...". Was this "all" including all inorganic salts and PMF-OA factors? If yes, I do not suggest saying, "the fact that they consist of various compounds with a wide range of volatilities", then I prefer to say, "the fact that they include various compounds with a wide range of volatilities".

Page 8, lines 4-7: I suggest the authors to separate this long sentence to be clearer.

Page 8, lines 7-11: Again, please separate this too much long sentence to be clearer.

Page 8, lines 12-14: Just as an example to separate a long sentence, ";" can be changed to ".".

Page 8, lines 16-18: I don't think that Jimenez et al., (2009) stated such conclusion. The authors could try to see variations of both total mass spectra and organic spectra, respectively, at different TD temperatures. The mass fraction of total NR-PM1 and total PMF-OA could be shown also.

Page 9, lines 4-5: May replace "classes/species" by "species". And replace "the total PM1 composition" by "the total PM1 mass loading".

Page 9, lines 5-7: Duplicate definition for "positive matrix factorization (PMF)". I don't think this sentence is useful here, as the authors said, "as discussed in section 2.4.". Please remove or reword it.

Page 9, lines 7-9: Same issue, this kind of information has been shown before in "2.4

Source Apportionment Method". Please reword.

Page 9, lines 9-11: Same again, this information has been shown in "2.4 Source Apportionment Method". And Duplicate definition for the abbreviation. Please reword.

Page 9, lines 11-13: How are the relative contributions of them at different TD-temperature conditions? The authors could be able to show that.

Page 9, lines 14-16: Please separate this sentence mixed with different information. For example, the authors could discuss the characteristics of your HOA mass spectrum, and give supporting reference. Then to compare your H/C value with typical ones as published, to further prove your reasonable HOA factor.

Page 9, line 16: Duplicate definition for "Black carbon (BC)".

Page 9, lines 16-17: Please provide reference(s) to support this discussion.

Page 9, lines 17-19: "as identified by previous publications (Zhang et al., 2007a; Lanz et al., 2007; Ulbrich et al., 2009)" seems not really needed here, as compared to the last sentence. It will be more useful to compare HOA with BC from traffic emissions (as I proposed above), because biomass burning emissions can also contribute BC here. Also, this will be helpful to support the Page 9, lines 17-19.

Page 9, line 23: Related reference(s) is(are) needed to be at the end of "...which are mainly ionized from alkanes, alkenes and, possibly, long chain fatty acids...".

Page 9, lines 23-24: "COA is characterized" can be removed. As presented "which are mainly ionized from alkanes, alkenes and, possibly, long chain fatty acids", what are such m/z 41 and m/z 55 from for the COA factor?

Page 9, lines 25-27: I did not get the meaning by mentioning this sentence about the results of Mohr et al. (2012). The authors even did not discuss your results about ratio of "m/z 55 to m/z 57".

Page 10, line 3: What are they different between "biomass burning" and "wood burning", as showing here together?

Page 10, lines 7-9: Reword this sentence.

Page 10, lines 10-12: I did not understand the relationship between "an O/C of 0.32 and showed a similar diurnal trend" and "indicating the significant influence...". If the authors would like to highlight the significant role of biomass burning emissions in aerosol pollution, you should provide the relative contribution to the PM loading. And please reword line 12.

Page 10, lines 13-16: Please reword this sentence. Separate it. And don't repeat to define LO-OOA and MO-OOA as shown before already.

Page 10, lines 16-18: Please separate this sentence. And provide the evidence of both sulfate and MO-OOA from regional transports.

Page 10, lines 20-21: How to get this conclusion of "denoting their secondary nature." only according to "...higher concentrations during the daytime,". Please explain it more.

Page 12, line 5: I feel like that almost findings relative to the OA volatility in this section "were consistent with" previous studies. For example, lines 9-10 (HOA), line 12 (BBOA), line 18 (COA), page 13 line 5 (OOA). I am not saying that the authors should not compare your findings with previous ones. But, the authors should find something new or that may improve our understanding. In addition, a couple of sentences are needed to be reworded/separated. E.g., some long sentences with ";" and with many times of "which attributive clause", etc.

Page 13, lines 8-17, and page 14, lines 13-15: As many previous studies, I do trust that some POA emissions are semivolatile, which might be a missing source of secondary organic aerosols. However, the authors did not provide direct and enough evidence that may support the conclusion of "HOA, rather than BBOA or COA, could be a potentially important source of LO-OOA" via "the oxidizing process of "Evaporation – Oxidation in gas phase - Condensation", although according to only the volatility sequence of the

PMF-OA factors.

---

## Author Comment (AC1) · 16 Nov 2017

This manuscript by Cao et al. presents the field measurement results using an AMS coupled with a TD conducted in China. As the application of TD-AMS is for now still rare in the atmospheric chemistry studies in China. The manuscript contains useful and valuable information. However, this reviewer finds that in the current version, many necessary details are missing, and many statements and important conclusions appear to be over-interpreted, not well supported by the data or at least is not well explained, which limits the scientific value of the work. The current version needs major revision before its acceptance. The current version fails to make good use of the wealthy data

obtained by the HR-AMS, and it should be extensively expanded. Here below I list the comments below that need to be addressed. Please pay attentions to the in-text citation format Reply: All corrected.

In the introduction part, please provide more details regarding previous TD-AMS studies, focused the important findings. Reply: New review of previous findings has been added into the introduction part as below: "A thermo-denuder (TD) is a device that is widely used to estimate aerosol volatility distributions (Wehner et al., 2002; An et al., 2007; Huffman et al., 2008; Xu et al., 2016). The TDs designed by Burtscher et al., 2001 and Wehner et al., 2002 are typically operated under temperatures higher than 200 °C and have average residence times from 0.3 to 9 s, focusing on very low volatility species. An et al., 2007 and Huffman et al., 2008 developed TDs with longer residence times to make them more suitable for measuring the volatility of semi-volatile organic aerosols. The combined TD and Aerodyne Aerosol Mass Spectrometer (TD-AMS) system was firstly applied in ambient study by Huffman et al. (2008) to quickly characterize the volatility of chemically-resolved ambient aerosol in a field campaign, and the temperature profiles, particle losses and key factors affecting the results were discussed. Huffman et al. (2009a) then measured the volatility of OA from different sources, including biomass-burning OA, meat-cooking OA, trash-burning OA, and chamber SOA formed from $\alpha$-pinene and gasoline vapours, and found semi-volatility for all the OAs, which is opposite to the previous atmospheric models that only regarded POAs as non-volatile species. Huffman et al. (2009b) also analyzed the positive matrix factorization (PMF) results based on the TD-AMS data and demonstrated that all types of OA should be regarded as semi-volatile species in the models. Lee et al. (2010) measured the volatility of aerosols with two different residence time sets and suggested that longer residence time was required to constrain the variation of OA volatility at lower concentrations. Obviously, OA volatilities, especially for different OA types, are still quite uncertain and need more ambient measurements to constrain."

(3) Introduction: Recently, Xu et al. (Atmos. Chem. Phys. 2014, 14, 12593-12611),

and Li et al. (Atmos. Environ. 2017, 158, 270-304.) have provided a summary of the AMS studies in China, these two papers should be cited. Reply: They are now cited.

(4) Section 2.2: The authors tested the transmission efficiencies, do these numbers are included to calculate the MFR? Besides, some theoretical calculations by Chris Cappa et al. regarding TD kinetics should be included. Reply: Yes, the transmission efficiencies were included to calculate the MFR, which has been clarified in the revised text. The model in Cappa et al. (2010) assumed many parameters and theoretical enthalpy to calculate the MFR, while our paper is currently focused on the experimental determination of MFR. Thus, the complex modeling of MFR will be considered in our future study.

Cappa, C. D.: A model of aerosol evaporation kinetics in a thermodenuder, Atmos. Meas. Tech., 3, 579-592, https://doi.org/10.5194/amt-3-579-2010, 2010.

(5) Section 2.2: why not to try a higher TD temperature, for example 250C or even 300C, there can be extremely low volatility species remained at such temperatures Reply: There are two main reasons for our temperature selection: 1. We referred to Huffman et al. (2008) for the setting of the range of the TD temperatures. 2. Only four temperature levels can be set in the software, which does not allow setting more temperature steps, and we care more about the relatively lower temperatures. It will be interesting to set the TD temperature up to 250 or 300 °C, and we would do it in future studies.

(6) Section 2: Some details regarding the TD is not provided. For example, what is the temperature cycle. What is the sampling time at each TD temperature? Do the authors change the absorption materials of the TD? And how often? Reply: The relevant information has been added as below: "The configuration of temperatures was: 35 min at 50 °C, 5 min for the temperature increasing to 100 °C, 22 min at 100°C, 5 min for the temperature increasing to 150 °C, 24 min at 150°C, 5 min for the temperature increasing to 200 °C, 25 min at 200 °C, and then 15 min for the temperature decreasing

to 50 °C. The complete temperature cycle was about 136 min." "The AMS was set with 4 menus: ByPass path in V-mode, TD path in V-mode, TD path in W-mode and ByPass path in W-mode, with 2 min in each menu. Only the data sampled during the stable temperature plateau (1839 points for V mode and 1842 points for W mode in TD-path, respectively) were selected for the calculation of volatility". We didn't change the absorption materials because the TD used was a rather new one in this campaign.

(7) Section 2: A lot of details are not provided. You should give a bit more details regarding the instrumental and operational details about the AMS, for the convenience of readers not in the AMS community. Calibrations? Relative ionization efficiencies? Size calibration? What are the chemical resolution? As your PMF includes ions with m/z up to 170, does these ions can be well separated from their neighborhood ions? Any instrument details about AE31? Reply: More details of the AMS and AE-31 have been added into the paper, as below. In our data processing, the high resolution of the W mode made the ions separated properly from their neighborhood ions when fitting the ions. "The AMS was set into two sampling ion optical modes: the V mode with UMR(unite mass resolution) was used for quantification of the UMR mass concentration and size distribution of the non-refractory species (including organics, sulfate, nitrate, ammonium, and chloride); the W mode was used to obtain the high-resolution mass spectra ($\sim$3000 m/$\Delta$m). The calibrations were conducted at the beginning and end of the campaign with the method described previously (Jayne et al., 2000; Jimenez et al., 2003; Drewnick et al., 2005), including the inlet flow rate, ionization efficiency calibration (IE), and particle size calibrations. The relative ionization efficiencies (RIE) used in the study were 1.2 for sulfate, 1.1 for nitrate, 1.3 for chloride, 1.4 for organics, and 4.0 for ammonium, respectively (Jimenez et al., 2003)." "An aethalometer (AE-31, Magee, US) coupled with a PM2.5 cyclone was used to measure the mass concentration of black carbon (BC) with a time resolution of 5 min. The wavelength of 880 nm was used to calculate the BC mass concentration in the data processing."

(8) Section 2.3: As this study are the very first few ones that used combined TD and

Bypass data to perform the PMF analyses, more details should be provided. The determination of optimal solution, what are the profiles of other solutions? What are the diagnostic plots of the currently chosen solution? What are the correlations of the factors with external tracers? what the PMF results are if only consider bypass data, and then to show clearly what are the advantages (please provide such information in the forms of plots in details not only a brief description) by running PMF on the combined TD and Bypass data. Such details are missing, but they are very helpful and also necessary to evaluate the robustness of your PMF results. Reply: In the supplement, the following plots are now presented: the profiles/time-series/diurnal variations/pie charts of the 4- and 6-factor solutions with combined TD and Bypass data; the diagnostic plots of the currently chosen solution; the PMF results of the profiles and time series of the 4-, 5-, and 6- factor solutions that only used the bypass data. The correlations of the factors with external tracers were also given. Relevant discussions are as below: "The solutions with more than five factors showed no distinct information but splitting of the factors; the Q/Qexpected showed the lowest value at fpeak=0; the varied fpeak did not improve the results; and the varied value of seed also made no significant difference of the solution. Therefore, the solution of five factors, fpeak = 0 and seed = 0, was determined as the optimal solution for this experiment..." "Compared to the results including the thermally denuded data, the HOA and OOA were mixed to some extent, with a signature of the high fraction of $CO_2^+$ in the HOA mass spectrum (Figure S2)."

(9) Results: P5L20 It is not convincing to state that sulfate is due to regional transport while other species are influenced by local emissions, only based on the extents of data variability (and the variability can be quantified, from xx to xx and what are the standard deviations?) Reply: This statement has been rephrased to be more rigorous, as below: "Sulfate showed a relatively stable time series, with a relative standard deviation (RSD) of 38.8%, compared to the other species, such as organics (RSD=56.1%), nitrate (RSD=69.6%), and black carbon (RSD=70.2%), indicating that sulfate was less affected by local emission sources."

[Figure]

(10) P5L25: the BC contribution seems to be quite high in PM1 (12.2%), is this number consistent with previous values in Shenzhen? Is this due to heavy traffic? That's also why I asked the calibration and operational details of AE31. Reply: The BC fraction in PM1 in November, 2009 was 14.0% in Shenzhen (He et al., 2011), which is even higher than that in this study. Actually, the high proportion of BC has been a feature of fine aerosol particles in Shenzhen, due to the high amounts of heavy-duty vehicles and ship emissions in this coastal city, with one of the top container ports in the world.

(11) P5L25: Is this composition for bypass conditions or others? Please indicate it clearly. Reply: It's only for the bypass conditions. We have made it clear.

(12) P5L29-: Please add the meteorological parameters, such as temperatures, solar radiation data to help explain the variations of nitrate. For example, it is not convincing to state the photochemical oxidation of NOx only based on the peak after BC peak. Similar for chloride, NH4NO3 is also semi-volatile, so why G/P partitioning drives chloride but not nitrate? Also, if you have PBL data, then add its diurnal pattern as well. Reply: The diurnal variation of temperature (which is available) is added into Figure 2b, and we have re-edited the discussion of the diurnal variation of nitrate, as below: "Nitrate showed a significant peak about 2 hours after the morning peak of BC, which was likely a result of photochemical oxidization of NOx emitted from the morning traffic. Then, the concentration of nitrate decreased because of both the lifting of the planetary boundary layer (PBL) and its evaporation at higher ambient temperatures (also shown in Figure 2b). Nitrate maintained at a stable concentration level in the evening."

(13) P6L7: Why not provide the neutralization plot? You can do that to support your statement. Reply: The neutralization plot has been added in the supporting information now. Relevant discussion has also been added as below: "The measured and predicted ammonium showed a similar correlation (R2=0.96–0.97) with a similar slope of 0.84–0.85 for both the ambient temperatures and 50 °C, implying that the aerosols showed some acidity in the real ambient temperature range (Zhang et al., 2007b)."

[Figure]

(14) P6L12: Do the elemental ratios calculated by Aiken 2008 method, or Canagaratna 2015 method? Please explain. Also, if it is from Canagaratna 2015 method, can it compare with previous values fairly? Reply: The elemental ratios were calculated based on the Canagaratna 2015 method, which has been clarified now in the text, as below. "The data analysis was performed with SQUIRREL 1.57 and PIKA 1.16 (http://cires1.colorado.edu/jimenez-group/ToFAMSResources/ToFSoftware/index.html) with the method in DeCarlo et al. (2006). The mass concentration was corrected with composition-dependent collection efficiency (Middlebrook et al., 2012). All the elemental ratios calculated here were based on the Improved–Ambient (I-A) method (Canagaratna et al., 2015), while the previous Aiken–Ambient (A-A) method was also used for comparison in Table S1 in the supporting information (Aiken et al., 2008)."

(15) P6L23: I don't think "which may be caused by the internal mixing and similar gas-particle distribution processes" is correct. If this is correct, then why you state previously that nitrate is due to photochemistry while chloride is due to G/P partitioning? They are not consistent. Reply: This incorrect statement is now removed.

(16) P6L25: "products of photochemical reactions of VOCs have a significant influence on the organic pollution" I don't think your data presented till here can indicate this. And why only photochemical reactions of VOCs, and other possible processes are excluded? Reply: We agree that this statement may be ambiguous, and it has been re-analyzed as below: "Compared to other species, the peak of organics was slightly smaller, which was a result of the much broader size distribution of organics towards smaller sizes. This character of organic size distribution implies that urban fresh primary emissions contributed significantly to organic aerosol (Canagaratna et al., 2004; He et al., 2011)."

(17) P6L27: why a large size would mean a more aged sulfate? and why it would then mean it had been transported regionally? Reply: The sentence is now rephrased as below: "The peak of sulfate was slightly larger than the other species, suggesting sul-

fate was mostly associated with larger particles that had grown through gas to particle conversion and coagulation processes during air mass transport (Zhang et al., 2005; Huang et al., 2008)."

(18) P7L6: Is your AE31 downstream of the TD? If so, can you plot the MFR of BC measured by AE31? You didn't mention the instrument setup of AE31 at all. Reply: Only the AMS was downstream of the TD. The AE-31 was only used for ambient sampling, which has been clarified in the revised manuscript.

(19) P8L7: "sulphuric acid can be distinguished with temperature between 30 and 125 $^{\circ}$C", what does this mean? I don't quite understand. Reply: Sentence rephrased as: "sulfuric acid would evaporate under temperatures of 30–125 $^{\circ}$C, while ammonium sulfate and bisulfate would evaporate between 125 to 175 $^{\circ}$C."

(20) P8L10: this paragraph is not well written. You should provide the average OA MS at different TD temperatures to give more explanations. You didn't explain why a more rapid increase occurred from 150-200C, and "OA is heated and oxidized into more aged compounds", there is lacking any supporting evidence. Jimenez et al., 2009 does not say so. "the results is consistent with the OA evolution process (Jimenez et al., 2009)?" Please be more specific and not so vague, this paper doesn't have any TD-AMS results, although it does show that the O/C ratio of OA increases with the decrease of saturation of OA. Reply: Jimenez et al. (2009) is removed. The discussion has been re-edited, as below: "When examining the organic mass spectra at difference temperatures (in Figure S6), the elevation of O/C with temperature increasing was found to be reasonably related to increasing of CO2+. The O/C variation should be attributed to loss of more volatile species at lower temperatures, especially after 150 $^{\circ}$C. Previous PMF results usually correlated higher volatility with reduced species and lower volatility with more oxygenated species (Ng et al., 2010; Huang et al., 2012). In this study, the elevation of O/C with temperature increasing was closely related to the evaporation of more reduced organic components, as the PMF results indicated later."

(21) P9L15: In my viewpoint, your BC and HOA diurnal patterns don't have clear matched with traffic times. It is overall high during evening to noon, and low during afternoon, more relevant to PBL dynamics, likely. Reply: Yes, we agree that PBL also took effects on the diurnal pattern significantly, and we have mentioned the roles of both PBL and traffics in the revised sentences.

(22) P9L23: Your writing logistic is strange. Sometimes you mentioned the results of previous studies, normally, the readers expect to see what your data show, but instead you didn't mention them. Here, you mention the findings of Mohr et al., 2012 regarding m/z 55 and 57, then how does them behave in your MS? If you are not going to describe them for your data, then why you mention Mohr's results? Their results don't support your explanation. Reply: We have supplemented the description of our data, as below: "In this study, COA showed much more $C_3H_3O^+$ than HOA, and the ratio of m/z 55 to m/z 57 showed values larger than 2, indicating the origin of cooking emissions."

(23) P9L28: That's what I am talking about, you mention PBL influence on COA, why not on HOA, etc. Reply: PBL has influence on all the species, and we have added the description of PBL influence on HOA accordingly.

(24) P10L5: Do you have data of external BB tracer to use? Reply: In the revised manuscript, we have made BC source apportionment based on its light absorption spectrum, as suggested by another reviewer. Thus, we got BC from traffic (BCtr) and BC from biomass burning (BCbb). Thus, an external tracer, BCbb, is used for BBOA, and good correlation is also found, as shown in Figure 4b.

(25) P10L15: What are the correlation coefficients of LOOOA with sulfate, as well as MOOOA with nitrate? Typically, the important criteria is to compare the correlations coefficients, and then postulate the assignment of LOOOA and MOOOA (although it is not always the case.) It seems like LOOOA might have a better correlation with sulfate, while MOOOA is better correlated with nitrate from Figure 4b, how to explain? What are the cross correlation coefficients between diurnal patterns of these species? A more

comprehensive discussion regarding these SOA should be combined with meteorological data and also gaseous species, if available. Reply: For MO-OOA, its correlation with sulfate (R2=0.64) was much higher than with nitrate (R2=0.27), which is a typical result. For LO-OOA, its correlation with sulfate (R2=0.59) was indeed a little bigger than with nitrate (R2=0.46), which could be a combined result of the precursors, formation mechanisms, as well as volatility of LO-OOA. The above explanation has been added into the revised text. In terms of the cross correlation between the diurnal patterns, not any good correlation (R2=0.01-0.15) was found due to the limited data points (only for 24 hours). The analysis between OOA and meteorological and gaseous data was ever done, but no helpful information was obtained due to the complexity of formation of OOA, which is not the most important focus of this study.

(26) P10L20: When you talk about the mass contributions of different OA factors, do you mean the average contributions over all data including bypass and TD? Or it is only for the case of bypass conditions? You didn't indicate this clearly at all throughout the manuscript. Reply: It's only for the case of bypass conditions throughout the paper. We have clarified it in the relevant figure captions.

(27) P12Section 3.4: Can you show the mass fractions of different OA factors at different TD temperatures? Reply: This figure is added as Figure 5, and the following discussion is also added: "Figure 5 showed the mass fractions of the five factors at different TD temperatures. It is found that when the temperature increasing, the fraction of MO-OOA quickly increased up to 67.6% at 200 °C, while LO-OOA showed a reverse trend, accounting for only 2.9% at 200 °C, indicating that they had quite different volatilities. For HOA, COA, and BBOA, they also exhibited different volatilities, with HOA accounting for only 5-7% above 100°C, while the fraction of COA did not change much with the temperature increasing. The different volatilities of different OA factors will be discussed in more detail in the following section."

(28) P12Section3, 4: As I suggested before, this section doesn't discuss any compositional changes of OA, but only mention the variations of MFR. You have an HR-AMS

which can provide chemical information in very details, which is in fact the unique advantage of HRAMS, why not do so? Reply: We have added the compositional changes as in the reply to the above question. Utilizing the advantages of HR-MS, we can obtain the elemental composition change of OA (O/C, H/C, and N/C) with the temperature increasing, which is now added into section 3.2, and the new discussion is as below: "The relationship of the O/C, H/C, and N/C ratios with the TD temperature is also shown in Figure 3. It can be seen that O/C kept increasing as the temperature increased, especially after 150 °C, which is consistent with previous studies (Xu et al., 2016a). When examining the organic mass spectra at difference temperatures (in Figure S6), the elevation of O/C with temperature increasing was found to be reasonably related to increasing of $CO_2+$. Previous PMF results usually correlated higher volatility with reduced species and lower volatility with more oxygenated species (Ng et al., 2010; Huang et al., 2012). In this study, the elevation of O/C with temperature increasing was closely related to the evaporation of more reduced organic components, as the PMF results indicated later. On the other hand, H/C showed a reasonable reverse trend relative to that of O/C. N/C generally had an increasing trend with the TD temperature increasing, but N/C varied largely at the different TD temperatures, suggesting that the volatilities of N-containing compounds are complex."

(29) P13L10: It is not that convincing to state that HOA rather than BBOA or COA was a more important source of LO-OOA, based on only the volatility sequence. The sequence may change at different temperatures, then the conclusion will change according to the logistic used here. To prove this, molecular level analyses are required. For example, a HOA tracer compound get oxidized, and the products appear in LO-OOA. Also, heterogeneous or condensed-phase reactions may also produce LO-OOA related species, it is not definitive that the reactions have to proceed via evaporation-gas-phase oxidation-condensation. Such statement is high speculative without any solid evidence in my opinion. Reply: We have removed this speculation and have made a more cautious statement as below: "It should also be noted that, as the most volatile species in this study, HOA could be evaporated easily and thus have a larger

potential to experience the "evaporation–oxidation in gas phase–condensation" process, forming SOA (e.g., LO-OOA) as described in Huang et al. (2012). This potential can be further supported by the fact that it is difficult to resolve HOA in downwind regions far from urban and industrial areas in China (Huang et al., 2011; Zhu et al., 2016). Other studies also showed that semi-volatile hydrocarbons from diesel exhaust (Robinson et al., 2007) and crude oil (de Gouw et al., 2011) can be easily oxidized to SOA. Therefore, the modelling work needs to consider the process from HOA to SOA in future."

(30) P13: Basically, I think it is dangerous to state the evolution process of different OA factors based on only volatility data. Even you are lacking of other data, you can perform more analyses, such as the temporal variations, compositional changes, correlations with other species, and discuss these in more details. This can be done for a few cases. Reply: We have made a new discussion for this part, with the addition of the comparison of O/C and volatility for different OA factors, as below: "Figure 7 compares both the volatilities (MFR at 50 °C) and the O/C ratios of the five factors. The sequence of the volatilities can be summarized as HOA > LO-OOA > COA ≈ BBOA > MO-OOA. It can be easily found that the sequence of the volatilities of the OA factors does not completely follow the sequence of the O/C ratios. For example, although LO-OOA has a higher O/C ratio than BBOA and COA, LO-OOA is also more volatile (or with a lower MFR) than BBOA and COA. This clearly indicates the volatility of the OA factors depends not only on the oxygenation of organic compounds, but also other factors, e.g., molecular weight and mixing state. HOA is identified as the most volatile OA factor while MO-OOA is nearly non-volatile near the real atmospheric temperatures in Shenzhen, which is consistent with the results observed in Mexico and Paris (Cappa and Jimenez, 2010; Paciga et al., 2016). However, LO-OOA is the second volatile OA factor after HOA in Shenzhen, which is different from that in Mexico, where BBOA is more volatile than LO-OOA. Actually, the volatility of the aerosols directly from biomass burning have been measured to be quite variable, with an evaporation rate of 0.2–1.6%Åů°C-1, depending on the kinds of wood and combustion conditions (Huffman et

al., 2009a). The relatively lower volatility of COA was also identified in previous studies and attributed to the abundant fatty acids of low volatility in COA (Mohr et al., 2009; Paciga et al., 2016). Hong et al. (2017) recently reported the estimation of the organic aerosol volatility in a boreal forest in Finland using two independent methods, including using a VTDMA with a kinetic evaporation model and applying PMF to HR-AMS data. Semi-volatile and low-volatility organic mass fractions were determined by both methods, similar to our study in China. This implies that MO-OOA and LO-OOA, with different volatilities, could be popular organic aerosol components across the world. Hong et al. (2017) also pointed out that determining of extremely low volatility organic aerosols from AMS data using the PMF analysis should be explored in future studies."
* * *

---

## Author Comment (AC2) · 16 Nov 2017

This manuscript describes the volatilities of the PM1 chemical components by using the Thermo-Denuder – Aerosol Mass Spectrometer (TD-AMS) system, along with the positive matrix factorization (PMF) analysis. The results make some very important implications on the atmospheric chemistry of aerosol particles, as there appears to be the first report about such study under a polluted environment in China. Overall, the content of this study fits within the scope of ACP. I agree that this campaign was well-designed. However, the authors need to consider making more further clarifications/ evidences to support a couple of ambiguous discussion and/or conclusions in

this paper. Numerous corrections on the text editing are needed, including reference and abbreviation formats as well as language issues, etc. Please the authors carefully check that throughout the manuscript. Therefore, a major revision is needed before it would be accepted in ACP.

General comments: When I finished reading this manuscript, I feel like that the authors did not fully analyze such comprehensive data set, then, highlight the new findings during the discussion of this study, because I found a lot of "consistent comparison" between this study and previous studies. I can understand that the authors would like to support your results/ discussion/conclusions, and I am not saying that you should not do that. But the authors should try to find something new as those comparisons with previous works. More analysis could also be done to understand such data set. For example, for both TD-path and non-TD-path data: temporal variations (PM1 species and PMF-OA factors)? chemical changes under the different environment conditions? any evidence for potential origins of changing volatilities for these species (e.g., what's difference between marine and continental air masses)? Variations of size distributions, rather than averaged ones? etc. More details of experimental materials need to be shown wherever in the main text or supplementary. What's the duration for TD-path data? It's easier to understand for readers if the authors could show that, for instance, in the time series of Figure 2a. What's the time resolution of your measurements during the campaign? How did the authors calibrate the AMS, and what were the results, e.g., values of IE, RIENH4, RIESO4? The authors should show one figure for the relationship between measured NH4 and predicted NH4 for TD-path and non-TD-path data, respectively. I am not convinced by the state of a finding about ": : :that HOA, rather than BBOA or COA, could be a potentially important source of LO-OOA: : :", as shown in the abstract and the main text elsewhere, just based on current TD-AMS-PMF results. The authors should perform more analysis to support that. For example, typical cases analysis? Since the authors have the data of seven-wavelength light absorption, it will be useful to support your PMF-POA factors by performing source apportionment of black carbon (BC) with aethalometer data (Elser et al., 2016). Reply: All revised as

described in the reply to the detailed questions below.

Comments/suggestions in details:

(1). Please note that abbreviations should be used in the same format throughout all the manuscript. For example, page 1, line 12: "a TD-AMS (Thermo-Denuder – Aerosol Mass Spectrometer)" and page 1, line 20: "a hydrocarbon-like OA (HOA, : : :)". For the consistency, the authors may replace "a TD-AMS (Thermo-Denuder – Aerosol Mass Spectrometer)" by "a Thermo-Denuder – Aerosol Mass Spectrometer (TD-AMS)". Somewhere else if the same issue should also be done. Reply: All similar issues have been corrected throughout the paper.

(2). Please define abbreviations when using it for the first time. For instance, Page 1, line 13: submicron particulate matter (PM1); page 1, line 19: positive matrix factorization (PMF). Somewhere else if the same issue should also be done. Reply: All similar issues have been corrected throughout the paper.

(3). Page 2, lines 15-25, the authors should also introduce more about the major findings reported by those previous studies. Then, the authors may tell readers the missing knowledge according to the new findings of your study. Reply: New review of previous findings has been added into the introduction part as below: "A thermo-denuder (TD) is a device that is widely used to estimate aerosol volatility distributions (Wehner et al., 2002; An et al., 2007; Huffman et al., 2008; Xu et al., 2016). The TDs designed by Burtscher et al. (2001) and Wehner et al. (2002) are typically operated under temperatures higher than 200 °C and have average residence times from 0.3 to 9 s, focusing on very low volatility species. An et al. (2007) and Huffman et al. (2008) developed TDs with longer residence times to make them more suitable for measuring the volatility of semi-volatile organic aerosols. The combined TD and Aerodyne Aerosol Mass Spectrometer (TD-AMS) system was firstly applied in ambient study by Huffman et al. (2008) to quickly characterize the volatility of chemically-resolved ambient aerosol in a field campaign, and the temperature profiles, particle losses and key factors affecting

the results were discussed. Huffman et al. (2009a) then measured the volatility of OA from different sources, including biomass-burning OA, meat-cooking OA, trash-burning OA, and chamber SOA formed from $\alpha$-pinene and gasoline vapours, and found semi-volatility for all the OAs, which is opposite to the previous atmospheric models that only regarded POAs as non-volatile species. Huffman et al. (2009b) also analyzed the positive matrix factorization (PMF) results based on the TD-AMS data and demonstrated that all types of OA should be regarded as semi-volatile species in the models. Lee et al. (2010) measured the volatility of aerosols with two different residence time sets and suggested that longer residence time was required to constrain the variation of OA volatility at lower concentrations. Obviously, OA volatilities, especially for different OA types, are still quite uncertain and need more ambient measurements to constrain."

(4). Page 2, lines 29-31, I cannot understand the relationship of this sentence with the major story of this introduction. Reply: This sentence is now deleted.

(5). Page 3, line 15, Duplicate definition for "thermo-denuder (TD)", it has been defined in the first time in Page 2 line 15. In addition, abbreviation should be followed hereafter when it has been defined at the first time. The authors should carefully check the similar issues as others, e.g., black carbon, organic aerosol, etc. Reply: All similar issues have been corrected throughout the paper.

(6). Page 5, lines 9-15, it's hard to read these sentences Please re-edit. The authors may introduce your data treatment procedure, then/at the same time, you could give the reference(s) to support yours as well as explain why. Reply: Re-edited as the following: "In addition, the PMF results with the data obtained only under ambient temperatures were also explored and the best solution was presented in Figure S2 in the supplement. Compared to the results including the thermally denuded data, the HOA and OOA were mixed to some extent, with a signature of the high fraction of $CO_2+$ in the HOA mass spectrum (Figure S2). Therefore, the PMF solution with the inclusion of the thermally denuded data was confirmed as the final results for later discussion. Huffman et al. (2009b) also suggested that the PMF solution of all data collected both

with and without TD-processing could facilitate the separation of different OA factors by enhancing the contrast of the time series of these factors."

(7). Page 5, line 23, the AE-31 should be described in the experimental method section. More details should be also given, e.g., cutoff, and which wavelength you used for the equivalent black carbon concentration. It is same for SMPS in line 29, for which more description should be given in this section too. Reply: Information added as below: "An aethalometer (AE-31, Magee, US) coupled with a PM2.5 cyclone was used to measure the mass concentration of black carbon (BC) with a time resolution of 5 min. The wavelength of 880 nm was used to calculate the BC mass concentration in the data processing. A scanning mobility particle sizer (SMPS, TSI Inc.) was used to measure the particle number size distribution (mobility diameter: 15–600 nm) with a time resolution of 5 min. By assuming the densities of the components obtained in the literature (Kuwata et al., 2012; Poulain et al., 2014; Hu et al., 2017), the corresponding mass concentration can be calculated from the particle number size distribution."

(8). Page 5, lines 24-25, ": : :due to rain: : :", to state this, the authors should provide related rain data to prove it. And what's the link of ": : :sulfate showed a relatively stable: : :" to this "rain case" in this sentence? and I don't understand why the relatively stable time series of sulfate can be considered as regional transportation? The authors may perform more analysis on chemical species along with your ground-measured meteorological parameters. Also, for instance, air mass trajectory analysis would be also useful to help figure it out. Reply: We didn't get the rain data, but we took sampling notes if there was obvious precipitation events. The vague statement about the indication of regional transport was removed. Since the focus of this paper is to characterize aerosol volatility, section 3.1 actually serves as the background information of the sampling campaign, and we would not extend the discussion much about the relationship between the chemical species and meteorology. The relevant sentences have been rephrased as below: "Sulfate showed a relatively stable time series, with a relative standard deviation (RSD) of 38.8%, compared to the other species, such as organics (RSD=56.1%), nitrate (RSD=69.6%), and black carbon (RSD=70.2%), indicating that sulfate was less affected by local emission sources. However, all the species decreased their concentrations largely during January 12–13 due to a heavy rain event."

(9). Page 5, lines 23-30: It's hard to read such long sentence. The authors should separate it for each information what you want to discuss. Such kind of long sentences is also frequently showing somewhere in this paper. The authors should keep the similar modification. Reply: Sentences rephrased.

(10). Page 6, lines 2-3: The authors should make the plot to support this discussion. And it would be also interesting to see what's the different ratio of measured and predicted NH4 from TD-path and non-TD-path data. Reply: The plots of the measured and predicted NH4+ data with and without the TD have been added in the supporting information. The relative sentences have been rephrased as below: "The measured and predicted ammonium showed a similar correlation (R2=0.96–0.97) with a similar slope of 0.84–0.85 for both the ambient temperatures and 50 °C, implying that the aerosols showed some acidity in the real ambient temperature range (Zhang et al., 2007b)."

(11). Page 6, line 4: Double meanings between "diurnal variation" and "during the day" in one sentence. Please reword it and somewhere same is also needed. Reply: Sentences re-edited.

(12). Page 6, line 5: Please the authors provide any evidence to prove the contribution of "the activity of heavy duty vehicles" to BC in the evening. If it's a case, and what's the difference sources that contribute BC particles between morning and evening peaks? Indeed, I feel more like that biomass burning emissions (according to the next discussion of BBOA variations) might also contribute the evening peak of BC. That's also one of reasons that I propose the authors to perform the BC source apportionment. Reply: Following this comment, we did the BC source apportionment using the method in San-dradewi et al. (2008), and the sentences have been re-edited according to the results as below: "The two peaks in the diurnal variation of BC obviously match the traffic rush
hours at approximately 8:00 in the morning and the activities of heavy duty vehicles in the evening. When BC source apportionment was applied for our BC dataset with the method in Sandradewi et al. (2008), the results indicated that biomass burning-emitted BC also made a small contribution to the evening peak of BC (Figure S4)."

(13). Page 6, lines 6-10: The authors should be careful to state the nitrate variations just according to such diurnal peaks between BC and nitrate. For example, how did the authors indicate that the peak of nitrate after the BC one should be linked to photo-chemistry, and that the peak at around 14:00 is due to "the enhancement of sunlight"? And, why there was no influence of gas-particle partitioning on nitrate, as discussed only for chloride? Reply: The analysis of the diurnal variation of nitrate has been im-proved as below: "Nitrate showed a significant peak about 2 hours after the morning peak of BC, which was likely a result of photochemical oxidization of NOx emitted from the morning traffic. Then, the concentration of nitrate decreased because of both the lifting of the planetary boundary layer (PBL) and its evaporation at higher ambient tem-peratures (also shown in Figure 2b). Nitrate maintained at a stable concentration level in the evening."

(14). Page 6, lines 10-11: The authors should provide/link your evidence or any pub-lished work(s) to prove such kind of discussions. In the manuscript, somewhere else with the similar issue should be modified too. Reply: Sentences modified as below: "Similar to ammonium nitrate, ammonium chloride is also quite semi-volatile as re-vealed in section 3.2. Therefore, its diurnal variation was largely influenced by the ambient temperature, as well as the height of the PBL. Also according to section 3.2, sulfate is a less-volatile species and thus would not lose significant particulate mass when the ambient temperature increases."

(15). Page 6, lines 11-13: Remove "during the day". I cannot understand that ": : :a regional product of oxidation by SO2 that is transported ...", since I did not see the transported evidence of sulfate in this study. The authors may further analyze the temporal variations along with the size distribution of sulfate, and considering meteorological influence. Reply: "during the day" removed; the sentences have been modified to be more reasonable, as below: "As a secondary species from SO2 oxidation, sulfate showed a slight diurnal variation, indicating that it was less affected by the variation of the PBL. This implies that sulfate was not a typical ground-emitted species and could be better mixed in the PBL. Actually, aerosol sulfate in Shenzhen has been proved to be a species mostly from regional air mass transport (He et al., 2011; Huang et al., 2014)."

(16). Page 6, lines 14-15: Please provide the neutralization plot. It seems an odd sentence for ": : :so the diurnal variation of ammonium was influenced by sulfate, nitrate and chloride". It's generally true that ammonium measured by the aerodyne AMS is mainly in the form of ammonium sulfate, ammonium nitrate, and/or ammonium chloride. I do more trust that diurnal variations of ammonium can be also affected by such inorganic salts formation processes, besides other factors, e.g., atmospheric physical processes. So, the authors should reword it. Reply: The neutralization plot is added in the supporting information (Figure S5). The sentence has been reworded as below: "Since ammonium exists mostly in the forms of (NH4)2SO4, NH4NO3 and NH4Cl, its diurnal variation should be significantly affected by the formation processes of all these inorganic salts, besides atmospheric physical processes and semi-volatility."

(17). Page 6, lines 15-16: I don't think the authors need to repeat such information of organic aerosols as already provided in the introduction before (page 2 lines 13-14). Introduced before. Again, somewhere else, such kind of discussion, at least, the authors should provide reference(s) to support it. I suggest the authors to reedit and combine this sentence with the next one (lines 16-18). Reply: Suggestion taken. The sentence has been re-edited with reference supporting and combined with the next sentence, as below: "The diurnal variation of organics showed more fluctuation and a few peaks, consistent with its complex origins, e.g., vehicles, biomass burning, and secondary formation (He et al., 2011; Elser et al., 2016), which will be discussed in detail in section 3.3."

(18). Page 6, lines 19-21: It's complicate to read here with a lot of comparison in only one sentence. Were all the averaged reference values only from Chen et al. (2015)? In addition, why did not the authors compare those values of O/C and H/C with some results observed under other polluted environments of China? It might make sense to understand such knowledge over the regional scale for developing countries in Asia (e.g., China). Reply: Yes, all the averaged reference values are from Chen et al. (2015). Following the suggestion, we now only compare the O/C values with other polluted environments in China, as below: "The average values of O/C and H/C of organic aerosol were 0.52 and 1.61, respectively. The average O/C value in this campaign is within the typical O/C range of 0.28–0.56 previously observed under polluted urban environments in China (Huang et al., 2011, 2012; He et al., 2011; Xu et al., 2016b; Hu et al., 2016; Lee et al., 2013)."

(19). Page 6, lines 21-25: The authors did not explain those diurnal variations of O/C and H/C. And why only a small H/C peak at noon was discussed, but no explanation at the peak during the nighttime? Do the authors think the biomass burning could also influence H/C variations, in addition to traffic and cooking emissions? Reply: The discussion has been re-edited as below: "The diurnal variation of O/C plotted in Figure 2c shows elevated values during the daytime, which is a clear indicator of the formation of secondary organic aerosol with more oxygen, while H/C reasonably showed an opposite diurnal trend, with decreased values during the daytime. The quick elevation of H/C in the evening should be a combined result of various primary emissions, e.g., traffic, cooking, and biomass burning, which is supported by the source apportionment results discussed in section 3.3."

(20). Page 6, line 26: The authors should avoid highlighting "non-refractory species measured by the AMS" too many times over the manuscript, because readers will know that after you explain it at the first time (except for the special case). Please the authors carefully check that elsewhere. Reply: All corrected.

(21). Page 6, lines 26-28: Be careful making the conclusion of averaged "approximately 500 – 700 nm in the accumulation modes" linking to "all the species" being aged particles. For example, were "all the species" including primary emissions, as below discussed HOA and BBOA, as well as fresh OOA? In addition, the authors already discussed that nitrate can be formed just after the morning traffic rush hours, so is this also included? To understand so, the authors can do the time series of size distribution of each chemical species (including both inorganic aerosols and PMF-OA factors) instead of showing here. Reply: We have re-edited the sentences to avoid indicating all species are aged, as below. Since this manuscript focuses on aerosol volatility rather than size distribution, we will not extend the discussion of size distribution more. "The peaks of all the species were at approximately 500 – 700 nm in the accumulation modes, while organics apparently had more mass distribution at smaller sizes down to ~100 nm."

(22). Page 6, line 30: I don't understand why "a similar average size distribution" of these inorganic species is because of this. Reply: This vague statement is now removed.

(23). Page 6, lines 30-32: As reported the comment of 21, the authors should provide the size distribution of PMF-OA factors to prove this. In addition, how to prove "products of photochemical reactions of VOCs have a significant influence on the organic pollution.", while rather than other formation processes? Reply: Since PMF only works on the bulk OA mass spectrum data, it will not produce size distribution of OA factors. To be more rigorous, this statement has been re-edited as below: "Compared to other species, the peak of organics was slightly smaller, which was a result of the much broader size distribution of organics towards smaller sizes. This character of organic size distribution implies that urban fresh primary emissions contributed significantly to organic aerosol (Canagaratna et al., 2004; He et al., 2011)."

(24). Page 7, lines 1-2: How to understand here, the large size of sulfate being aged and from regional transports? Reply: The sentence is now rephrased as below: "The peak of sulfate was slightly larger than the other species, suggesting sulfate was mostly

associated with larger particles that had grown through gas to particle conversion and coagulation processes during air mass transport (Zhang et al., 2005; Huang et al., 2008)."

(25). Page 7, line 8: Again, "measured by the AMS", such kind of words, does not need to be iterate. Reply: All corrected.

(26). Page 7, line 10: What does mean by "measured directly by the AMS."? Is it meaning the measured particles from non-TD-path channel? Reply: Yes. Now it is reworded as "The MFR is calculated as the ratio of the species mass concentrations with and without TD-processing."

(27). Page 7, lines 12-13: The authors stated ": : :of the total non-refractory species and organics all: : :". Was this "all" including all inorganic salts and PMF-OA factors? If yes, I do not suggest saying, "the fact that they consist of various compounds with a wide range of volatilities", then I prefer to say, "the fact that they include various compounds with a wide range of volatilities". Reply: Suggestion taken. The relative sentence is corrected to "the fact that they include various compounds with a wide range of volatilities".

(28). Page 8, lines 4-7: I suggest the authors to separate this long sentence to be clearer. Page 8, lines 7-11: Again, please separate this too much long sentence to be clearer. Page 8, lines 12-14: Just as an example to separate a long sentence, ";" can be changed to ".". Reply: All corrected.

(29). Page 8, lines 16-18: I don't think that Jimenez et al., (2009) stated such conclusion. The authors could try to see variations of both total mass spectra and organic spectra, respectively, at different TD temperatures. The mass fraction of total NR-PM1 and total PMF-OA could be shown also. Reply: Jimenez et al. (2009) is removed. New discussion has been edited, as below: "When examining the organic mass spectra at difference temperatures (in Figure S6), the elevation of O/C with temperature increasing was found to be reasonably related to increasing of $CO_2^+$. The O/C variation should

be attributed to loss of more volatile species at lower temperatures, especially after 150 °C. Previous PMF results usually correlated higher volatility with reduced species and lower volatility with more oxygenated species (Ng et al., 2010; Huang et al., 2012). In this study, the elevation of O/C with temperature increasing was closely related to the evaporation of more reduced organic components, as the PMF results indicated later."

(30). Page 9, lines 4-5: May replace "classes/species" by "species". And replace "the total PM1 composition" by "the total PM1 mass loading". Reply: Corrected.

(31). Page 9, lines 5-7: Duplicate definition for "positive matrix factorization (PMF)". I don't think this sentence is useful here, as the authors said, "as discussed in section 2.4.". Please remove or reword it. Reply: Sentence removed.

(32). Page 9, lines 7-9: Same issue, this kind of information has been shown before in "2.4 Source Apportionment Method". Please reword. Page 9, lines 9-11: Same again, this information has been shown in "2.4 Source Apportionment Method". And Duplicate definition for the abbreviation. Please reword. Reply: The sentences have been reworded as below: "As discussed in section 2.4, PMF modelling was applied to the high-resolution mass spectra of organics and five factors were identified with their MS profiles shown in Figure 4a. Under ambient temperatures, HOA, COA, BBOA, LO-OOA, and MO-OOA averagely accounted for 13.5%, 20.6%, 8.9%, 39.1%, and 17.9% of the total organic mass, respectively (Figure 4d)."

(33). Page 9, lines 11-13: How are the relative contributions of them at different TD temperature conditions? The authors could be able to show that. Reply: This figure is added as Figure 5, and the following discussion is also added: "Figure 5 showed the mass fractions of the five factors at different TD temperatures. It is found that when the temperature increasing, the fraction of MO-OOA quickly increased up to 67.6% at 200 °C, while LO-OOA showed a reverse trend, accounting for only 2.9% at 200 °C, indicating that they had quite different volatilities. For HOA, COA, and BBOA, they also exhibited different volatilities, with HOA accounting for only 5-7% above 100°C, while

the fraction of COA did not change much with the temperature increasing. The different volatilities of different OA factors will be discussed in more detail in the following section."

(34). Page 9, lines 14-16: Please separate this sentence mixed with different information. For example, the authors could discuss the characteristics of your HOA mass spectrum, and give supporting reference. Then to compare your H/C value with typical ones as published, to further prove your reasonable HOA factor. Reply: The sentence is re-edited as below: "HOA is most often dominated by long chain hydrocarbon ion series of $C_n H_{(2n+1)}^+$ and $C_n H_{(2n-1)}^+$ in previous findings (Canagaratna et al., 2004; Mohr et al., 2009; Ng et al., 2010), which is also the case in this campaign. The average O/C of HOA was 0.10 in this campaign, which was in its range (0.03 to 0.17) reported in previous publications (e.g., Aiken et al., 2009; Huang et al., 2010; Mohr et al., 2012)."

(35). Page 9, line 16: Duplicate definition for "Black carbon (BC)". Reply: Corrected.

(36). Page 9, lines 16-17: Please provide reference(s) to support this discussion. Page 9, lines 17-19: "as identified by previous publications (Zhang et al., 2007a; Lanz et al., 2007; Ulbrich et al., 2009)" seems not really needed here, as compared to the last sentence. It will be more useful to compare HOA with BC from traffic emissions (as I proposed above), because biomass burning emissions can also contribute BC here. Also, this will be helpful to support the Page 9, lines 17-19. Reply: Following the suggestion, we made BC source apportionment and compare HOA to BC from traffic to support the discussion. The sentences are reworded as below: "BC is regarded as a tracer of HOA, and can be significantly emitted from both fossil fuel combustion and biomass burning (Zhang et al., 2007a; Lanz et al., 2007; Lan et al., 2011). The good correlation (R2=0.82) of HOA and BC from traffic (Figure 4b) suggested that HOA was mainly from traffic emissions. The diurnal variation of HOA was influenced by PBL dynamics and also showed peaks that matched the rush hours, further supporting the dominant role of traffic emissions to the HOA."

(37). Page 9, line 23: Related reference(s) is(are) needed to be at the end of ": : :which are mainly ionized from alkanes, alkenes and, possibly, long chain fatty acids: : :". Reply: References added with (He et al., 2010; Huang et al., 2010; Mohr et al., 2009; Mohr et al., 2012).

(38). Page 9, lines 23-24: "COA is characterized" can be removed. As presented "which are mainly ionized from alkanes, alkenes and, possibly, long chain fatty acids", what are such m/z 41 and m/z 55 from for the COA factor? Reply: This sentence is now removed since its information is duplicate to the next sentence.

(39). Page 9, lines 25-27: I did not get the meaning by mentioning this sentence about the results of Mohr et al. (2012). The authors even did not discuss your results about ratio of "m/z 55 to m/z 57". Reply: The new sentence has been added as below: "In this study, COA showed much more $C_3H_3O^+$ than HOA, and the ratio of m/z 55 to m/z 57 showed values larger than 2, indicating the origin of cooking emissions."

(39). Page 10, line 3: What are they different between "biomass burning" and "wood burning", as showing here together? Reply: It's a mistake. "wood burning and" is removed.

(40). Page 10, lines 7-9: Reword this sentence. Page 10, lines 10-12: I did not understand the relationship between "an O/C of 0.32 and showed a similar diurnal trend" and "indicating the significant influence: : :". If the authors would like to highlight the significant role of biomass burning emissions in aerosol pollution, you should provide the relative contribution to the PM loading. And please reword line 12. Reply: The sentences have been reworded as below: "The O/C ratios of BBOA varied a lot in previous studies. Laboratory studies reported O/C ratios of 0.18–0.26 for six types of biomass burning emissions (He et al., 2010), and O/C ratios of 0.31 for lodgepole pine burning and 0.42 for sage/rabbitbrush burning (Aiken et al., 2008). Decarlo et al. (2010) reported an O/C ratio of 0.42 for ambient biomass burning aerosol. The BBOA in this study showed an O/C ratio of 0.33, which is within the range of previous studies. The

diurnal trend of BBOA showed a large peak in the evening, well consistent with the diurnal peak of BC from biomass burning in Figure S4."

(41). Page 10, lines 13-16: Please reword this sentence. Separate it. And don't repeat to define LO-OOA and MO-OOA as shown before already. Reply: Corrected

(42). Page 10, lines 16-18: Please separate this sentence. And provide the evidence of both sulfate and MO-OOA from regional transports. Reply: The sentence has been rephrased and citations have been added to support, as below: "The factor with a relatively higher O/C ratio (0.95) of OOA and higher f44 than f43 is identified as MO-OOA. It showed a good correlation (R2=0.64) with sulfate, which was less volatile and had been identified as a regional pollutant in Shenzhen (He et al., 2011; Huang et al., 2014), implying MO-OOA could also be aged aerosol from regional transport."

(43). Page 10, lines 20-21: How to get this conclusion of "denoting their secondary nature." only according to ": : :higher concentrations during the daytime,". Please explain it more. Reply: The sentence has been modified as below: "Unlike the primary organic components, which had lower concentrations during the daytime due to the elevated PBL, the diurnal variations of both LO-OOA and MO-OOA showed higher concentrations during the daytime, suggesting that photochemical secondary production should be their main source."

(44). Page 12, line 5: I feel like that almost findings relative to the OA volatility in this section "were consistent with" previous studies. For example, lines 9-10 (HOA), line 12 (BBOA), line 18 (COA), page 13 line 5 (OOA). I am not saying that the authors should not compare your findings with previous ones. But, the authors should find something new or that may improve our understanding. In addition, a couple of sentences are needed to be reworded/separated. E.g., some long sentences with ";" and with many times of "which attributive clause", etc. Reply: We have made a new discussion for this part, highlighting the features and significance of our study, as below: "Figure 7 compares both the volatilities (MFR at 50 °C) and the O/C ratios of the five factors. The

sequence of the volatilities can be summarized as HOA > LO-OOA > COA ≈ BBOA > MO-OOA. It can be easily found that the sequence of the volatilities of the OA factors does not completely follow the sequence of the O/C ratios. For example, although LO-OOA has a higher O/C ratio than BBOA and COA, LO-OOA is also more volatile (or with a lower MFR) than BBOA and COA. This clearly indicates the volatility of the OA factors depends not only on the oxygenation of organic compounds, but also other factors, e.g., molecular weight and mixing state. HOA is identified as the most volatile OA factor while MO-OOA is nearly non-volatile near the real atmospheric temperatures in Shenzhen, which is consistent with the results observed in Mexico and Paris (Cappa and Jimenez, 2010; Paciga et al., 2016). However, LO-OOA is the second volatile OA factor after HOA in Shenzhen, which is different from that in Mexico, where BBOA is more volatile than LO-OOA. Actually, the volatility of the aerosols directly from biomass burning have been measured to be quite variable, with an evaporation rate of 0.2–1.6%·°C-1, depending on the kinds of wood and combustion conditions (Huffman et al., 2009a). The relatively lower volatility of COA was also identified in previous studies and attributed to the abundant fatty acids of low volatility in COA (Mohr et al., 2009; Paciga et al., 2016). Hong et al. (2017) recently reported the estimation of the organic aerosol volatility in a boreal forest in Finland using two independent methods, including using a VTDMA with a kinetic evaporation model and applying PMF to HR-AMS data. Semi-volatile and low-volatility organic mass fractions were determined by both methods, similar to our study in China. This implies that MO-OOA and LO-OOA, with different volatilities, could be popular organic aerosol components across the world. Hong et al. (2017) also pointed out that determining of extremely low volatility organic aerosols from AMS data using the PMF analysis should be explored in future studies."

(45). Page 13, lines 8-17, and page 14, lines 13-15: As many previous studies, I do trust that some POA emissions are semivolatile, which might be a missing source of secondary organic aerosols. However, the authors did not provide direct and enough evidence that may support the conclusion of "HOA, rather than BBOA or COA, could

be a potentially important source of LO-OOA" via "the oxidizing process of "Evaporation – Oxidation in gas phase - Condensation", although according to only the volatility sequence of the PMF-OA factors. Reply: We have removed this speculation and have made a more cautious statement as below: "It should also be noted that, as the most volatile species in this study, HOA could be evaporated easily and thus have a larger potential to experience the "evaporation–oxidation in gas phase–condensation" process, forming SOA (e.g., LO-OOA) as described in Huang et al. (2012). This potential can be further supported by the fact that it is difficult to resolve HOA in downwind regions far from urban and industrial areas in China (Huang et al., 2011; Zhu et al., 2016). Other studies also showed that semi-volatile hydrocarbons from diesel exhaust (Robinson et al., 2007) and crude oil (de Gouw et al., 2011) can be easily oxidized to SOA. Therefore, the modelling work needs to consider the process from HOA to SOA in future."

---

## Author Comment (AC3) · 16 Nov 2017

In this study, the authors deployed a TD-AMS to investigate the volatility of different chemical compositions of PM1 in Shenzhen. Aerosol volatility studies are important but rare in China. This work, as far as I know, could be the 1st report on the online measurement of the volatilities of aerosol chemical components using a TD-AMS in China, and thus provide valuable information. In addition, OA was classified into several groups using PMF and tested their volatility separately, which help to understand their sources and characters. The manuscript is overall well written and documented. The topic fits well in the scope of ACP. I recommend this manuscript can be published

after some revisions.

1) In general, aerosol is a really complex system, especially in China. The analysis in this work is a bit too simple. Some more discussions are recommended, e.g. size revolved volatility study, organic aerosol volatility in different events, e.g. heave haze events or NPF events. Reply: According to this comments and other comments of this reviewer and other reviewers, we have added many new useful materials to support the data analysis in this paper, with the major revisions summarized below: (1) The elemental composition change of organic aerosol with TD temperature, as below: "The relationship of the O/C, H/C, and N/C ratios with the TD temperature is also shown in Figure 3. It can be seen that O/C kept increasing as the temperature increased, especially after 150 °C, which is consistent with previous studies (Xu et al., 2016a). When examining the organic mass spectra at difference temperatures (in Figure S6), the elevation of O/C with temperature increasing was found to be reasonably related to increasing of $CO_2^+$. Previous PMF results usually correlated higher volatility with reduced species and lower volatility with more oxygenated species (Ng et al., 2010; Huang et al., 2012). In this study, the elevation of O/C with temperature increasing was closely related to the evaporation of more reduced organic components, as the PMF results indicated later. On the other hand, H/C showed a reasonable reverse trend relative to that of O/C. N/C generally had an increasing trend with the TD temperature increasing, but N/C varied largely at the different TD temperatures, suggesting that the volatilities of N-containing compounds are complex." (2) The mass fractions of the five OA factors at different TD temperatures, with the discussion as below: "Figure 5 showed the mass fractions of the five factors at different TD temperatures. It is found that when the temperature increasing, the fraction of MO-OOA quickly increased up to 67.6% at 200 °C, while LO-OOA showed a reverse trend, accounting for only 2.9% at 200 °C, indicating that they had quite different volatilities. For HOA, COA, and BBOA, they also exhibited different volatilities, with HOA accounting for only 5-7% above 100°C, while the fraction of COA did not change much with the temperature increasing. The different volatilities of different OA factors will be discussed in more

detail in the following section." (3) The comparison between the volatility and the O/C ratios for the OA factors, with the new discussion as below: "Figure 7 compares both the volatilities (MFR at 50 °C) and the O/C ratios of the five factors. The sequence of the volatilities can be summarized as HOA > LO-OOA > COA $\approx$ BBOA > MO-OOA. It can be easily found that the sequence of the volatilities of the OA factors does not completely follow the sequence of the O/C ratios. For example, although LO-OOA has a higher O/C ratio than BBOA and COA, LO-OOA is also more volatile (or with a lower MFR) than BBOA and COA. This clearly indicates the volatility of the OA factors depends not only on the oxygenation of organic compounds, but also other factors, e.g., molecular weight and mixing state. HOA is identified as the most volatile OA factor while MO-OOA is nearly non-volatile near the real atmospheric temperatures in Shenzhen, which is consistent with the results observed in Mexico and Paris (Cappa and Jimenez, 2010; Paciga et al., 2016). However, LO-OOA is the second volatile OA factor after HOA in Shenzhen, which is different from that in Mexico, where BBOA is more volatile than LO-OOA. Actually, the volatility of the aerosols directly from biomass burning have been measured to be quite variable, with an evaporation rate of 0.2–1.6%·°C-1, depending on the kinds of wood and combustion conditions (Huffman et al., 2009a). The relatively lower volatility of COA was also identified in previous studies and attributed to the abundant fatty acids of low volatility in COA (Mohr et al., 2009; Paciga et al., 2016). Hong et al. (2017) recently reported the estimation of the organic aerosol volatility in a boreal forest in Finland using two independent methods, including using a VTDMA with a kinetic evaporation model and applying PMF to HR-AMS data. Semi-volatile and low-volatility organic mass fractions were determined by both methods, similar to our study in China. This implies that MO-OOA and LO-OOA, with different volatilities, could be popular organic aerosol components across the world. Hong et al. (2017) also pointed out that determining of extremely low volatility organic aerosols from AMS data using the PMF analysis should be explored in future studies."

2) The last sentence of conclusion part, "HOA, rather than BBOA or COA, could be

a potentially important source of LO-OOA". More discussions are needed to support this statement. Reply: We have made a more cautious statement as below: "It should also be noted that, as the most volatile species in this study, HOA could be evaporated easily and thus have a larger potential to experience the "evaporation–oxidation in gas phase–condensation" process, forming SOA (e.g., LO-OOA) as described in Huang et al. (2012). This potential can be further supported by the fact that it is difficult to resolve HOA in downwind regions far from urban and industrial areas in China (Huang et al., 2011; Zhu et al., 2016). Other studies also showed that semi-volatile hydrocarbons from diesel exhaust (Robinson et al., 2007) and crude oil (de Gouw et al., 2011) can be easily oxidized to SOA. Therefore, the modelling work needs to consider the process from HOA to SOA in future."

3) Hong et al., 2017 has reported a similar work that estimate of the organic aerosol volatility using two independent methods including a VTDMA and HR-AMS. They compared the direct measurement result from VTDMA and PMF result from HR-AMS. It would be good to add some discussions to compare the methods and results between this two works. Hong, J., Äijälä, M., Häme, S. A. K., Hao, L., Duplissy, J., Heikkinen, L. M., Nie, W., Mikkilä, J., Kulmala, M., Prisle, N. L., Virtanen, A., Ehn, M., Paasonen, P., Worsnop, D. R., Riipinen, I., Petäjä, T., and Kerminen, V. M.: Estimates of the organic aerosol volatility in a boreal forest using two independent methods, Atmos. Chem. Phys., 17, 4387-4399, 10.5194/acp-17-4387-2017, 2017. Reply: The discussion about Hong et al. (2017) has been added into the text as below: Hong et al. (2017) recently reported the estimation of the organic aerosol volatility in a boreal forest in Finland using two independent methods, including using a VTDMA with a kinetic evaporation model and applying PMF to HR-AMS data. Semi-volatile and low-volatility organic mass fractions were determined by both methods, similar to our study in China. This implies that MO-OOA and LO-OOA, with different volatilities, could be popular organic aerosol components across the world. Hong et al. (2017) also pointed out that determining of extremely low volatility organic aerosols from AMS data using the PMF analysis should be explored in future studies.

4) I suggest adding a summary of volatility studies in China. Although there should be no other studies using a TD-AMS, but some related work using VTMDA are still worth to be summarized, e.g. Cheung et al., 2016; Nie et al., 2017. Cheung, H. H. Y., Tan, H., Xu, H., Li, F., Wu, C., Yu, J. Z., and Chan, C. K.: Measurements of non-volatile aerosols with a VTDMA and their correlations with carbonaceous aerosols in Guangzhou, China, Atmos. Chem. Phys., 16, 8431-8446, 10.5194/acp-16-8431-2016, 2016. Nie, W., Hong, J., Häme, S. A. K., Ding, A., Li, Y., Yan, C., Hao, L., Mikkilä, J., Zheng, L., Xie, Y., Zhu, C., Xu, Z., Chi, X., Huang, X., Zhou, Y., Lin, P., Virtanen, A., Worsnop, D. R., Kulmala, M., Ehn, M., Yu, J. Z., Kerminen, V. M., and Petäjä, T.: Volatility of mixed atmospheric Humic-like Substances and ammonium sulfate particles, Atmos. Chem. Phys. Discuss., 2016, 1-26, 10.5194/acp-2016-839, 2016. Reply: The following sentences have been added into the introduction part: "Cheung et al. (2016) studied the aerosol volatility in Guangzhou, China, based on the volatility tandem differential mobility analyzer (VTDMA) measurement, and found that non-volatile organic aerosol may contribute significantly to the non-volatile residuals. Also using a VTDMA, Nie et al. (2017) studied the volatility of aerosol humic-like substances (HULIS) in Nanjing, China, and figured out that the interaction between HULIS and ammonium sulfate tended to decrease the volatility of organic aerosols."

---

## Author Response (AR2)

The authors basically addressed most of my comments, and it is very much improved with more details than the original version. I have a few remaining small questions before it can be accepted.

1)CsCl solution was atomized and used for the particle loss test? Any specific reason to use this solution?

Reply:

Yes, the CsCl solution was atomized and used for the particle loss test. As discussed in Huffman et al. (2008), CsCl was chosen as a standard chemical species to test the TD response as it showed virtually no evaporation with temperature increasing up to 250 ℃. This reference is cited now.

2)It is strange to me that the TD software only allows to set 4 temperatures, which is obviously not enough. But anyway, if this is the case, describe it in the main text.

Reply:

The case that we had utilized all the temperature settings was clarified in text.

3)A chemical resolution of 3000 is likely not high enough to resolve ions at m/z 170. I understand the separation of ions depends other factors not only chemical resolution, and since the ions at such high m/zs are likely very minor, so this is probably fine, but it is better to explain more.

Reply:

Though the ions at higher m/z cannot be separated as well as the ions at low m/z, the ions with m/z up to 170 showed very significant signals and can still be separated properly in our dataset. This has been clarified in the text.

4)"The peak of sulfate was slightly larger than the other species, suggesting sulfate was mostly associated with larger particles that had grown through gas to particle conversion and coagulation processes during air mass transport (Zhang et al., 2005; Huang et al., 2008)." Gas-to-particle conversion typically results in smaller particles (condensation mode), thus it does not suggest that sulfate associating with large particles had grown through GP conversion. This sentence still needs re-phrase.

Reply:

The sentence is re-edited as "The peak of sulfate was slightly larger than the other species, suggesting that sulfate was mostly associated with more aged particles that had grown during air mass transport (Zhang et al., 2005; Huang et al., 2008)."

Anonymous Referee #2

The authors addressed my comments from the first review step and revised the manuscript. However, I still have some comments that need to be addressed as the following areas.

1. For the framework of "3 Results and discussion", I feel like that the authors should consider re-organizing it. Why did not discuss "3.4 Volatility of OA factors" together with the section of "3.2 Volatility of PM1 species"? The "3.3 Source apportionment" section can be discussed before "3.2 Volatility of PM1 species", which may just follow "3.1 PM1 chemical compositions".

Reply:

These paragraphs are now re-ordered as suggested.

2. Is it possible that the authors could show the time series of the observed data with TD-processing?

Reply:

The time series of the five non-refractory species with TD processing were shown as below, but it seems not necessary to put it in the paper.

[Figure]

3. More detailed descriptions (e.g., show the calculation formulas, alpha values for fossil fuel and biomass burning, etc.) regarding to the source apportionment of black carbon need to be provided in main text or supplement. Otherwise, the readers would have a similar/same question about that.

Reply:

The following sentences are added into the supplement.

The total BC ($BC_{total}$) measured by the aethalometer can be separated into BC emitted by traffic ($BC_{tr}$) and biomass burning ($BC_{bb}$) with the aerosol absorption coefficients ($b_{abs}$) at 470 nm and 950 nm, as shown in the following equations discussed in Sandradewi et al. (2008):

$$\frac{b_{abs}(470nm)_{tr}}{b_{abs}(950nm)_{tr}} = \left(\frac{470}{950}\right)^{-\alpha_{tr}} \qquad (1)$$

$$\frac{b_{abs}(470nm)_{bb}}{b_{abs}(950nm)_{bb}} = \left(\frac{470}{950}\right)^{-\alpha_{bb}} \qquad (2)$$

$$b_{abs}(\lambda) = b_{abs}(\lambda)_{tr} + b_{abs}(\lambda)_{bb} \qquad (3)$$

$$BC_{tr} = BC_{total} \cdot \frac{b_{abs,tr,950nm}}{b_{abs,total,950nm}} \qquad (4)$$

$$BC_{bb} = BC_{total} - BC_{tr} \qquad (5)$$

Where $\alpha$ is the absorption exponent and $\lambda$ is the wavelength. The values of

$\alpha$ used for traffic and biomass burning are 0.9 and 1.7, respectively, according to the values used in Xi'an, China, by Elser et al. (2016).

4. Page 4 Lines 1-2: Does it means that all RIEs for each species were from default/reference values, as showing a reference of (Jimenez et al., 2003)? Can the authors please show your calibration results (figures) in supplement? My mention here is because some discussions relative to aerosol acidity that was resulting from the neutralization plot (Page 7 Lines 12-13). Atmospheric aerosol acidity can be largely affected by particulate sulfate and ammonium concentrations. This means if the RIE calibrations for sulfate and ammonium weren't performed well, then it's very hard to discuss any information about aerosol acidity.

Reply:

Yes, the RIEs used in this study were the default values shown in Jimenez et al. (2003). The RIE of sulfate wasn't calibrated since it is regarded to be stable in the previous studies. The relationship of the molar concentrations of nitrate and ammonium in the IE calibration is shown as below, and it showed a slope of 0.96, very close to 1.0, confirming that the RIE used for ammonium was reasonable.

[Figure]

5. Page 16 lines 23-24 and Page17 lines 1-2. How could we get "…fact that it is difficult to resolve HOA in …" that can support the previous discussion? Again, I did not find direct evidence that can prove "evaporation–oxidation in gas phase–condensation" process, even from Huang et al. (2012).

Reply:

The sentence has been supplemented with "…fact that it is difficult to resolve HOA in… implying that HOA had been oxidized during the air mass transport (Huang et al., 2011; Zhu et al., 2016)."

As to "the evaporation–oxidation in gas phase–condensation process", we stated that this is a possibility since HOA was found semi-volatile and previous studies showed that semi-volatile hydrocarbons can be easily oxidized in the gas phase to form less volatile SOA (Robinson et al., 2007; de Gouw et al., 2011). The relevant sentence has been rephrased to make this point clearer.

P3L89:

and the ions with m/z up to 170 could be separated properly in this study.

P4L100:

The full configuration of temperatures in the TD software was:

P4L116:

The total BC ($BC_{total}$) measured by the aethalometer can be separated into BC emitted by traffic ($BC_{tr}$) and biomass burning ($BC_{bb}$) based on the aerosol absorption as described in the supplement (Sandradewi et al., 2008).

P5L130:

CsCl was chosen as the standard chemical species for the test of the transmission efficiency of TD due to its thermal stability as discussed by Huffman et al. (2008).

P8L208:

The peak of sulfate was slightly larger than the other species, suggesting that sulfate was mostly associated with more aged particles that had grown during air mass transport (Zhang et al., 2005; Huang et al., 2008).

P16L354:

Huang et al. (2012) also pointed out that HOA could be oxidized to SOA based on the analysis of their diurnal variations.

P16L356:

[revised manuscript text omitted]